# Physicochemical Properties of Starch Binary Mixtures with *Cordia* and *Ziziphus* Gums

**Abdellatif Mohamed \*, Shahzad Hussain** **, Mohammed S. Alamri, Mohammed A. Ibraheem, Akram A. Abdo Qasem** **and Ibrahim A. Ababtain**

Department of Food Science and Nutrition, King Saud University, Riyadh 1145, Saudi Arabia; shhussain@ksu.edu.sa (S.H.); msalamri@ksu.edu.sa (M.S.A.); mfadol@ksu.edu.sa (M.A.I.); aqasem@ksu.edu.sa (A.A.A.Q.); ababtain.ibr@gmail.com (I.A.A.)
\* Correspondence: abdmohamed@ksu.edu.sa; Tel.: +966-11-4698546

**Abstract:** The effect of gum *Cordia* (GC) and gum *Ziziphus* (GZ) on the physicochemical properties of wheat, potato, and chickpea starches was investigated. Native or acetylated gums were mixed with starch at 2% or 5%. Starches were analyzed using rapid viscoanalyzer (RVA), differential scanning calorimeter (DSC), texture analyzer, and rheometer. In the presence of gums, the data showed clear variations between the starch gels. The effects of gum acetylation on the starch characteristics were significant. According to the starch type, the peak viscosity of the gels increased depending on the gum type or concentration. With the exception of the potato starch, when gums were added, the gelatinization temperature of the starches increased. Gum acetylation significantly increased starch–gel elasticity (high G′), particularly at the 2% concentration. GC-starch gel hardness was ranked as follows: chickpea–5% native gum > wheat–5% native gum > potato–0% gum, whereas GZ followed the order of: chickpea–2% native gum > wheat–2% native gum > potato–2% native gum. Both the gums promoted reduction in syneresis for the wheat and chickpea starches. Although there was no clear trend, the *Ea* of the native starches was lowered overall as a result of the gums, indicating the limited effect of temperature on the rheological properties of the blends.

**Keywords:** *Cordia*; *Ziziphus*; gums; starch; texture; rheology; pasting



## 1. Introduction

Gums are a sub-category of hydrocolloids and one of the most widely utilized soluble fibers in the food industry. They are mostly long chain polysaccharides that are hydrophilic with high molecular weights [1]. Their function in foods can be summarized as: water holding, thickening, gelling, stabilizing, film forming, as well as being texture modifier agents. For instance, some products use gums for the abovementioned functionalities, i.e., breads, cakes, mayonnaises, dressings, desserts jellies, and ice creams [2]. To achieve specific functionality, gums are incorporated singly or in combination. The addition of gums into a wheat flour mixture has the potential to improve the textural characteristics and freeze–thaw stability, besides serving as an anti-staling agent of the baked product during storage [3,4]. At sufficiently low concentrations, gums can give baked products rubbery crumbs and elastic texture, which may be perceived as softer or fresher but, at elevated concentrations, the product can be tough or chewy [5]. The addition of limited amounts of gums may also make low-fat or fat-free products available, as well as increase the overall soluble fiber content of the baked products [6].

Generally, gum sources include, plant exudate (gum Arabic), plant structure (pectin, cellulose), seeds (guar gum, locust bean gum), tuber (konjac mannan), algal (agar, carrageenan, alginate), and microbial-based sources such as xanthan gum, curdlan, and gellan [7]. Currently, the best known and most utilized plant-based hydrocolloids in the food industry include alginates, carrageenans, agar, guar gum, gum Arabic, and carboxymethyl

cellulose [8]. One of the relatively new sources of gum is *Cordia myxa* (lasura), which is a tree grown in tropical and sub-tropical environments. This tree produces fruits with a substantial amount of gum that can potentially be a plant-based gum. The shape of the *Cordia myxa* fruit is round to oval shaped drupes that are about 15–20 mm in diameter, are arranged in yellow to orange–pink in clusters, as shown in the picture, and has a sweet taste [9]. In Table 1, the chemical composition of *Cardia myxa* is presented.

**Table 1.** Proximate analysis of Cordia fruit [10].

| Component | % (per 100 g of Edible Portion) |
|---|---|
| carbohydrates | 57 |
| Crude proteins | 8.32 |
| Total sugars | 3.55 |
| Reducing sugars | 3.41 |
| Non reducing sugars | 0.08 |
| Pectin | 4.5 |
| **Minerals (mg/g) dry basis** | |
| Sodium | 1.62 |
| Potassium | 7.83 |
| Calcium zinc | 0.46 |
| Zinc | 0.35 |
| Iron | 0.51 |

The phytochemical screening of the fruit revealed the presence of alkaloids, anti-inflammatory agents, steroids, and polyphenols, which points to its great medicinal and nutritional value [11]. The fresh Cordia fruit contains a viscose transparent mass of polysaccharide that turns into a brown color when exposed to the air. The composition of the gummy mass includes anionic polysaccharides rich in uronic acid, glucose, rhamnose, xylose, and mannose [12].

Gum *Cordia* has pseudoplastic behavior with a high viscosity, which is suitable for use as a strong emulsifier, thickener, and stabilizer in the food and pharmaceutical industries [13]. Previous reports showed that gum *Cordia* had a significant effect on the physicochemical properties of apple jelly [14,15]. The total phenol content of the jelly sample significantly increased with the addition of the gums. The rheological properties showed that gel samples containing 75% gum *Cordia* were similar to the control, but had the highest apparent viscosity, loss moduli (G″), storage moduli (G′), and complex viscosity [14]. *Cordia* gum has an edible coating supplemented with $CaCl_2$ or ascorbic acid, which has a significant positive effect on weight loss, ascorbic acid content, and total phenolic compounds, as well as browning and inhibition of polyphenol oxidase activity [16]. Authors have previously published some of the successful utilization of gum *Cordia* as a coating material on nuts to retard oxidative rancidity and moisture loss [17–19].

*Zizyphus* Spina-Christi is a tree species belonging to the botanical family *Rhamnaceae* and is native to a vast area of Africa stretching from Mauritania through the Sahara and the coasts zones of West Africa to the Red Sea. It is drought resistant, very resistant to heat, and can be found in desert areas where groundwater is available. Reports in the literature indicated that the ethanol extract of *Ziziphus* fruits exhibited flowing properties similar to xanthan gum and higher than guar gum [20]. Optimum conditions for *Ziziphus* mucilage extraction were reported as 1:7 water, 60 °C, and 1:3 ethanol for precipitation. The water holding capacity, oil absorption, and emulsion capacity were 73.35 g water/g dry basis, 4.97 g oil/g dry sample, and 52.22%, respectively. This data showed that oil absorption was higher than guar gum and xanthan gum, but emulsion capacity was lower [21]. The shape, size, color, relative density, and seed weight of the fruits of most *Ziziphus* varieties

vary [22]. A total of five types of phenolic compounds were detected in Napek extract, a *Ziziphus* type, but only two were identified by TLC as caffeic acid and p-coumaric acid [23]. Many researchers are focused on finding new sources as well as establishing new uses of natural gums. The purpose of this study was to isolate polysaccharides from *Ziziphus* and *Cordia myxa* fruits and examine if these isolated polysaccharides can influence the physicochemical properties of starch from diverse sources (cereal, legume, or tuber), which will have a direct impact on starchy product formulation.

## 2. Materials and Methods

### 2.1. Materials

Chickpea and wheat were purchased from the local market (Riyadh, Saudi Arabia) and potato starch was purchased from Sigma Aldrich (St. Louis, MO, USA). *Cordia* and *Ziziphus* fruits were collected from a local farm in Riyadh, Saudi Arabia. Sodium hydroxide, acetic anhydride, and hydrochloric acid were purchased from Sigma Aldrich.

### 2.2. Chickpea Starch Isolation

Chickpea (CP) whole meal was slurried in distilled water (50:50) and blended by a heavy-duty blender for 5 min (BioloMix, Whirlpool corporation, Benton Harbor, MI, USA); it was then filtered through a 200-mesh sieve and the filtrate was centrifuged at $2000 \times g$ for 15 min (Fisherbrand™ Refrigerated Centrifuge GT2, Hamburg, Germany) [24]. Following centrifugation, the top layer on the precipitate was removed and the white pellet was re-suspended in distilled water and centrifuged as before. This process was repeated four times, after which a white pure starch fraction was obtained. The obtained starch was air-dried using acetone, ground in a coffee grinder, and stored at 4 °C.

### 2.3. Wheat Starch Isolation

Wheat starch isolation was done according to Alamri et al. [25]. Wheat flour dough (flour/water ratio 2:1, $w/w$) was prepared, placed in a cloth, kneaded, and washed thoroughly with excess water. The temperature of the water was 25 °C. The filtrate was then centrifuged at $2000 \times g$ for 20 min (Fisherbrand™ Refrigerated Centrifuge GT2, Hamburg, Germany) and the top dark layer on the precipitate was removed and the white material was re-suspended in distilled water as before. The isolated starch was air dried, ground in a coffee grinder, and stored at 4 °C.

### 2.4. Isolation of Gums

*Cardia* or *Ziziphus* fruits were destoned, thoroughly washed, and steamed for 3 min to inhibit the enzymatic browning. Pulp was prepared by blending the fruits at high-speed using an auxiliary kitchen mixer (BioloMix, Whirlpool corporation, Benton Harbor, MI, USA) then put in 25 °C distilled water at 1:3 ratios for 1 min, filtered through a muslin cloth, and centrifuged at $10,000 \times g$ for 30 min (Fisherbrand™ Refrigerated Centrifuge GT2). The supernatant was collected, neutralized, and freeze-dried (Alpha 1-4, LD plus, at 0.005 mBarr and $-50$ °C). It was then ground to powders, passed through a 60-mesh sieve, and stored in airtight jars at 4.0 °C for further use. The gum yields based on percentage of total fruit was 1.8% and 11.5% for GC and GZ, respectively.

### 2.5. Gums Modification

Acetylation was done by preparing a 10% ($w/v$) aqueous solution of freeze-dried GC or GZ powder (20 g/200 mL), which was then stirred for 60 min at 40 °C while the pH was maintained between 8–8.5 using 0.50 N sodium hydroxide. Acetic anhydride (4 mL) was added in drops while the pH was maintained between 8–8.5. At the end of the reaction, the pH of the contents was adjusted to 6.0–6.5 using 0.50 N HCl, and the final product was freeze-dried, passed through a 60-mesh sieve, and stored in an airtight container.

## 2.6. Determination of Degree of Acetylation

The percent degree of acetylation was determined by a simple titration method adopted from reference [26] with minor modifications. One gram (1.0 g) of acetylated gums was added to 50 mL of distilled water, stirred, and heated at 40 °C until complete dissolution, then the pH was recorded. After this, 40 mL of 0.5 M NaOH was added, stirred for 15 min, and the excess alkali was back titrated using 0.5 M HCl until the point of neutralization. As a blank, the native unmodified gum solution was prepared and treated as above. Acetyl (%) and degree of (DS) substitution was calculated as:

$$\% \text{ Acetyl} = \frac{[(blank\ mL - sample\ mL)] \times HCl\ molarity \times 0.043 \times 100}{sample\ weight\ (g)} \tag{1}$$

$$\text{Degree of substitution} = \frac{[(162) \times \%\ acetylation]}{[4300 - (42 \times \%\ acetylation)]} \tag{2}$$

The titration volume of the blank (untreated starch) and sample (treated starch) was in milliliters, whereas the sample weight was in grams. DS is defined as the average number of sites per glucose unit that possesses an acetyl group.

## 2.7. Preparation of Gum Starch Blends (Binary Composites)

A total weight of 25 g of blend was prepared by replacing 2% or 5% of the starch (wheat, potato, or chickpea) with gum (GC or GZ) and the blend was suspended in 50 mL of distilled water, mixed sorely, and freeze-dried to powder (Table 2).

**Table 2.** Details of the preparation of starch gum blends.

| Wheat Starch | 100% Wheat Starch |
|---|---|
| 2% native cordia | 2 g native cordia—98 g wheat starch |
| 2% acetylated cordia | 2 g acetylated cordia—98 g wheat starch |
| 5% native cordia | 5 g native cordia—95 g wheat starch |
| 5% acetylated cordia | 5 g acetylated cordia—95 g wheat starch |
| 2% native ziziphus | 2 g native ziziphus—98 g wheat starch |
| 2% acetylated ziziphus | 2 g acetylated ziziphus—98 g wheat starch |
| 5% native ziziphus | 5 g native ziziphus—95 g wheat starch |
| 5% acetylated ziziphus | 5 g acetylated ziziphus—95 g wheat starch |
| **Potato Starch** | **100% Potato Starch** |
| 2% native cordia | 2 g native cordia—98 g potato starch |
| 2% acetylated cordia | 2 g acetylated cordia—98 g potato starch |
| 5% native cordia | 5 g native cordia—95 g potato starch |
| 5% acetylated cordia | 5 g acetylated cordia—95 g potato starch |
| 2% native ziziphus | 2 g native ziziphus—98 g potato starch |
| 2% acetylated ziziphus | 2 g acetylated ziziphus—98 g potato starch |
| 5% native ziziphus | 5 g native ziziphus—95 g potato starch |
| 5% acetylated ziziphus | 5 g acetylated ziziphus—95 g potato starch |

**Table 2.** *Cont.*

| Chickpea Starch | 100% Chickpea Starch |
| --- | --- |
| 2% native cordia | 2 g native cordia—98 g chickpea starch |
| 2% acetylated cordia | 2 g acetylated cordia—98 g chickpea starch |
| 5% native cordia | 5 g native cordia—95 g chickpea starch |
| 5% acetylated cordia | 5 g acetylated cordia—95 g chickpea starch |
| 2% native ziziphus | 2 g native ziziphus—98 g chickpea starch |
| 2% acetylated ziziphus | 2 g acetylated ziziphus—98 g chickpea starch |
| 5% native ziziphus | 5 g native ziziphus—95 g chickpea starch |
| 5% acetylated ziziphus | 5 g acetylated ziziphus—95 g chickpea starch |

*2.8. Fourier Transform Infrared (FTIR) Analysis of Native and Acetylated Gums*

The spectra of the native and modified gums were obtained using an ATR-FTIR spectrophotometer (ALPHA ATR, Brukner, Germany). The samples were placed directly on the ATR diamond attachment. The ATR Spectral scanning was carried out with a scan speed of 1 cm/s at a resolution of 4 cm$^{-1}$. The analysis was done under transmittance (%) mode within the frequency range of 400–4000 cm$^{-1}$.

*2.9. Differential Scanning Calorimetry (DSC)*

DSC analysis on the individual starches and their blends was done using a DSC Q 2000 (TA Instruments, New Castle, DE, USA). A starch sample (6–8 mg) was weighed in aluminum pans with 60–80% ($w/w$) distilled water being added as well. Hermetically sealed pans were allowed to equilibrate at room temperature for 2 h and scanned from 30 to 120 °C at 10 °C/min. An empty pan was placed on the reference cell. The tested parameters of the thermal analysis were enthalpy $\Delta H$ (J/g), onset temperature ($T_o$), and peak temperature (PT), and were found using the Universal Analysis Software provided by the DSC manufacturer (TA Instruments, New Castle, DE, USA).

*2.10. Rapid Visco Analyzer Measurements (RVA)*

Starch/gum blends, or their corresponding controls (3 g at 14% moisture basis), were weighed directly into special RVA canisters and a total weight of 28 g was completed with distilled water. The control was considered starch only and no gum. The slurry was heated to 50 °C and held for 30 s and then was heated to 95 °C in 4.40 min (at 10.23 °C/min) and held for 4 min. The sample was allowed to cool to 50 °C in 4 min and was held at 50 °C for 2 min [25]. The profile of the tested samples includes the peak viscosity of the formed gel, final viscosity, setback, and pasting temperature.

*2.11. Freeze Thaw Stability of Starch Gels*

Starch gels were stored in centrifuge tubes at −20 °C. After 4 days, the frozen gels were allowed thaw in a water bath at 50 °C for 30 min, centrifuged at 3000× $g$ for 15 min at 10 °C, and the separated water was calculated. For another freeze–thaw cycle, the gels were re-stored in the freezer for another 4 days. Syneresis was calculated by subtracting the separated water after each cycle from the weight before the cycle. The separation of water from the gels after four or eight days of storage was reported as %Syneresis [24].

*2.12. Gel Texture*

The starch gels prepared in the RVA were transferred to a 25 mL beaker and stored overnight at room temperature. The gels were compressed using a TA-TXT texture analyzer cylinder (Vienna Court, Lammas Road, UK) at a speed of 0.5 mm/s to a distance of 10 mm in two penetration cycles. The measured gel characteristics include: hardness, cohesiveness, and gumminess, which were calculated according to reference [27].

### 2.13. Rheological Measurements and Temperature Dependence

Dynamic rheology measurements from a frequency sweep test at the angular frequency from 0.1–100 rad/s at a 5% strain were carried out on RVA-cooked starch samples using a DHR-Hybrid Rheometer (TA Instruments, New Castel, PA, USA). The experimental data included storage modulus (G′), loss modulus (G″), and dynamic mechanical loss tangent (tanδ = G″/G′). Steady shear behaviors (shear rate vs shear stress) and temperature dependence (30, 40 and 50 °C) of the cooked starch gels were recorded at a variable shear rate of 1 to 100/s. The data were fitted to the power law model (Equation (3)).

$$t = K\gamma^n \tag{3}$$

where, t = shear stress ($10\ \mathrm{dyn/cm^2}$), K = consistency coefficient ($\mathrm{Pa\ s^n}$), $\gamma$ = shear rate ($\mathrm{s^{-1}}$), and n = flow behavior index (dimensionless).

The temperature dependency was estimated by the Arrhenius Equation (Equation (4)) [28], which was applied to investigate the effect of temperature (30, 40, and 50 °C) on the flow consistency index at a shear rate of $100\ \mathrm{s^{-1}}$ for the starch gel:

$$\ln \mu = \ln A - \frac{Ea}{R}\frac{1}{T} \tag{4}$$

where, $\mu$ is the is the apparent viscosity (Pa s) at $100\ \mathrm{s^{-1}}$ at the measurement temperature ($\mathrm{Pas\ s^n}$), $A$ is the pre-exponential factor ($\mathrm{Pa\ s^n}$), $Ea$ is the activation energy (J/mol), $R$ is the universal gas constant ($8.314\ \mathrm{J\ mol^{-1}\ K^{-1}}$), and $T$ is the absolute temperature (Kelvins). The value of $Ea$ at each treatment was calculated from the regression analysis of $\ln \mu$ vs. $1/T$.

### 2.14. Statistical Analysis

The measurements were done in triplicate and the data were analyzed using ANOVA. A factorial design was applied to test for the effects of GC and GZ on starch. Duncan's multiple range test was applied to compare the means at $p \leq 0.05$ using the PASW® Statistics 18 software (SPSS Inc., Hong Kong, China).

## 3. Results and Discussion

### 3.1. FTIR

The FTIR spectra for both gums are shown in Figure 1. A typical absorption band at $3270\ \mathrm{cm^{-1}}$ indicates the presence of OH groups, whereas the band at $2922\ \mathrm{cm^{-1}}$ is a characteristic C-H stretching of polysaccharide sugars such as arabinose, galactose, and rhamnose [29]. The native gum exhibited a reduction in the OH band-intensity compared to both the acetylated gums, which can be attributed to the replacement of the OH with acetyl group. Furthermore, Figure 1 displays bands of varied intensities due to the added acetyl groups, i.e., the stretching at $1585–1559\ \mathrm{cm^{-1}}$, which is ascribed to the ester bond of the C=O group.

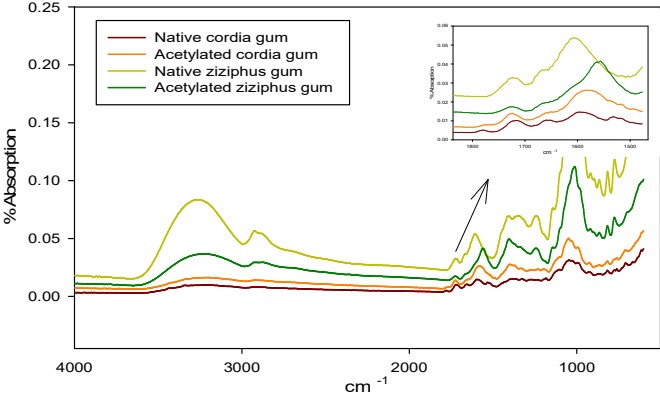

**Figure 1.** FTIR spectra of native and acetylated *Cordia* and *Ziziphus* gums.

### 3.2. Degree of Acetylation

The degree of acetylation for GC and GZ was 0.78 and 0.17, respectively. The hydroxyl groups of GC were substituted around five times more than those of GZ. This could be attributable to the fact that the hydroxyl groups in GC seem to be more readily accessible than in GZ. Given that the acetylated GC is much more hydrophobic than GZ, due to the additional acetyl groups, it can be employed in oil-based applications.

### 3.3. Differential Scanning Calorimetry (DSC)

The onset ($T_o$) and peak (PT) temperatures and the gelatinization enthalpy ($\Delta H$) of the starch and starch/gum blends were determined (Table 3). The effect of GC and GZ on the thermal characteristics of the investigated starches varied widely. The $T_o$ and PT of the wheat and chickpea starches increased significantly at a higher gum content, however, potato starch did not follow a particular pattern—low GC raised the $T_o$ and PT and greater GZ decreased it. The effect of high GC or GZ concentration on the $T_o$ and PT of the wheat and chickpea starches was significant, notably for the 5% native gum and the 5% acetylated gum. The 5% native GC or GZ considerably increased the $T_o$ and PT of the chickpea and wheat starches, whereas the 5% acetylated GC and 2% GZ greatly raised the $T_o$ and PT of the chickpea and wheat starches (Table 3). When compared to native gum, the 5% acetylated GZ gum appeared to reduce $T_o$ and PT of the wheat and potato starches (Table 3). Regardless of the gum content, the native GC raised the gelatinization enthalpy ($\Delta H$) of the wheat starch, while the acetylated GC decreased it, but both the native and acetylated GZ lowered it at both concentrations significantly. According to previous reports in the literature, acetylation can reduce the thermal characteristics of gums by lowering the melting enthalpy, peak, and onset temperatures [30]. This could explain why starch/gum blends containing acetylated GC or GZ gum had a lower $\Delta H$. Except for the 5% native GZ, the potato starch without gum showed a considerable reduction in the $\Delta H$ regardless of the gum type or concentration. This illustrates how the $\Delta H$ of the cereal and tuber starches can differ, emphasizing the significance of the starch source. Of course, the discrepancy is connected to the physical structure of the granules, which includes the amylose content and granule compactness, both of which have a direct impact on granule swelling at the start of the gelatinization process. Unlike the other two starches, the $\Delta H$ of the chickpea starch was greatly reduced by 2% native or 2% acetylated GC, and significantly increased by the 5% native GC. In general, as compared to the wheat and potato starch, the chickpea starch has different gelatinization characteristics, such as peak viscosity and setback, which could explain the differences in the $\Delta H$ measurements. Conversely, regardless of the concentration or type of modification, GZ significantly increased the $\Delta H$ of the chickpea starch (Table 3). As a result, the effect of the gums on the tested starches differed depending on whether the gum was native or acetylated in addition to the gum concentration. Water migration to the crystalline portion of the starch granule may have been delayed by the gums, resulting in an increase in the $\Delta H$. At the stages of gelatinization, strong hydrogen bonding between leached amylose and the gum may lead to poorer contact between the amylose and amylopectin, causing gelatinization to occur sooner, which could be attributed to the reduction in the $\Delta H$ of the potato starch by both gums [31]. However, the increase in the PT of the starches caused by the gums could be related to the gum's hydrophilicity, which enhances its ability to absorb and hold water. Water availability is crucial for starch swelling, which is the first step in the gelatinization process. Therefore, water holding can delay starch granule swelling, thus the entire gelatinization process, which is signified by a higher PT. This conclusion is consistent with reports in the literature [32,33]. Hence, the strength with which each gum holds water, which is controlled by its structural makeup or hydrophobicity, could account for the differences in the effect. With reference to the chickpea starch, the effects of GC and GZ were radically different, with GC significantly lowering the $\Delta H$, whereas GZ increased it. (Table 3). The level of contact between the gum and the leached amylose hypothesis could explain this discrepancy (mentioned above). As a result, when compared to GZ, there is, possibly, a greater interaction between the leached

amylose and GC, which could lead to faster gelatinization, but it could also be influenced by the length of the amylose chain.

**Table 3.** Thermal properties of wheat, potato, and chickpea starch gels containing different levels of *Cordia* and *Ziziphus* gums.

| | [1] ΔH (J/g) | [2] $T_0$ (°C) | [3] PT (°C) |
|---|---|---|---|
| **Wheat Starch** | | | |
| *Cordia* **gum** | | | |
| 0 | 9.34 ± 0.12 [b] | 55.35 ± 0.11 [d] | 62.59 ± 0.23 [d] |
| 2% native | 9.55 ± 0.10 [b] | 56.44 ± 0.16 [c] | 64.42 ± 0.27 [b] |
| 2% acetylated | 6.64 ± 0.06 [c] | 57.78 ± 0.26 [b] | 63.50 ± 0.06 [c] |
| 5% native | 10.09 ± 0.05 [a] | 58.72 ± 0.23 [a] | 65.61 ± 0.28 [a] |
| 5% acetylated | 5.56 ± 0.21 [d] | 58.84 ± 0.29 [a] | 65.43 ± 0.14 [a] |
| *Ziziphus* **gum** | | | |
| 0 | 9.34 ± 0.12 [a] | 55.35 ± 0.11 [c] | 62.59 ± 0.23 [bc] |
| 2% native | 6.04 ± 0.04 [c] | 56.09 ± 0.59 [bc] | 62.33 ± 0.31 [c] |
| 2% acetylated | 6.76 ± 0.45 [b] | 58.06 ± 0.14 [a] | 64.01 ± 0.26 [a] |
| 5% native | 5.49 ± 0.20 [c] | 58.12 ± 0.20 [a] | 63.54 ± 0.15 [a] |
| 5% acetylated | 4.83 ± 0.33 [d] | 56.51 ± 0.45 [b] | 62.90 ± 0.13 [b] |
| **Potato starch** | | | |
| *Cordia* **gum** | | | |
| 0 | 15.97 ± 0.15 a | 59.19 ± 0.04 [c] | 63.33 ± 0.07 [c] |
| 2% native | 15.39 ± 0.16 [b] | 61.14 ± 0.04 [a] | 65.83 ± 0.05 [a] |
| 2% acetylated | 14.33 ± 0.16 [c] | 58.96 ± 0.16 [c] | 62.80 ± 0.06 [d] |
| 5% native | 14.20 ± 0.13 [c] | 59.76 ± 0.07 [b] | 64.05 ± 0.14 [b] |
| 5% acetylated | 13.02 ± 0.07 [d] | 57.69 ± 0.26 [d] | 62.90 ± 0.19 [d] |
| *Ziziphus* **gum** | | | |
| 0 | 15.97 ± 0.15 [a] | 59.19 ± 0.04 [b] | 63.33 ± 0.07 [b] |
| 2% native | 16.30 ± 0.47 [a] | 57.57 ± 0.88 [c] | 62.42 ± 0.39 [c] |
| 2% acetylated | 11.33 ± 0.16 [b] | 55.44 ± 0.10 [d] | 60.85 ± 0.27 [d] |
| 5% native | 16.16 ± 0.10 [a] | 60.45 ± 0.08 [a] | 65.10 ± 0.24 [a] |
| 5% acetylated | 10.08 ± 0.39 [c] | 57.80 ± 0.34 [c] | 63.14 ± 0.30 [b] |
| **Chickpea starch** | | | |
| *Cordia* **gum** | | | |
| 0 | 4.78 ± 0.13 [a] | 54.87 ± 0.15 [e] | 63.04 ± 0.45 [c] |
| 2% native | 3.03 ± 0.05 [c] | 57.99 ± 0.21 [c] | 65.34 ± 0.11 [b] |
| 2% acetylated | 3.09 ± 0.10 [c] | 57.68 ± 0.15 [d] | 65.43 ± 0.27 [b] |
| 5% native | 5.00 ± 0.06 [a] | 59.23 ± 0.08 [b] | 66.78 ± 0.13 [a] |
| 5% acetylated | 3.53 ± 0.26 [b] | 59.59 ± 0.03 [a] | 66.94 ± 0.41 [a] |
| *Ziziphus* **gum** | | | |
| 0 | 4.78 ± 0.13 [c] | 54.87 ± 0.15 [d] | 63.04 ± 0.45 [c] |
| 2% native | 6.49 ± 0.23 [a] | 57.67 ± 0.16 [b] | 65.14 ± 0.13 [b] |
| 2% acetylated | 5.44 ± 0.15 [b] | 57.28 ± 0.03 [c] | 65.27 ± 0.04 [b] |
| 5% native | 5.79 ± 0.04 [b] | 59.34 ± 0.08 [a] | 66.03 ± 0.22 [a] |
| 5% acetylated | 5.46 ± 0.17 [b] | 59.41 ± 0.14 [a] | 66.20 ± 0.07 [a] |

[1] ΔH = gelatinization enthalpy; [2] $T_0$ = onset temperature; [3] PT = peak temperature; Mean carrying same letters in columns for particular gum under different starches are non-significantly different from each other.

### 3.4. Rapid Visco Analyzer (RVA)

The pasting properties of the tested starches are listed in Table 4. The gel prepared from the wheat starch/gum blends had a significant rise in peak viscosity (PV), however the gum type or concentration had a varying degree of effect on the gel. The PV of the wheat starch gel was increased by the 5% acetylated gum *Cordia* (GC) from 2970 to 3876 centipoise (cP), whereas the 5% native GC increased it from 2970 to 4172. Unlike the acetylated, the native GC had a concentration-dependent impact on the wheat starch, as a greater gum

enhanced the PV more (Table 4). In contrast, the 2% and 5% acetylated GZ increased the PV from 2970 to 3271 and 3162 (cP), respectively, while the 2% acetylated had the highest PV. The native GZ exhibited no concentration dependency, because the 2% and the 5% values were very close, while the native GC exhibited a concentration dependency. The increase in the PV by both gums indicates an increase in the total amount of absorbed water, which indicates greater starch granule swelling. In the case of the wheat starch, the level of the added acetylated GZ or GC had little effect on the PV, unlike the native gum (Table 4). The 5% native gum had the maximum PV of the wheat/GC blend, while the 2% acetylated gum had the highest for the wheat/GZ blend. When compared to the 2% acetylated GZ, the 5% native GC increased the PV of the wheat starch by 809 cP, whereas the 2% acetylated GZ improved the PV by 108 cP, indicating that GC is a better alternative than GZ when higher a PV is sought. The addition of either gums to the potato starch gel resulted in a considerable reduction in the PV, with the 2% acetylated GC and 2% native GZ having the least PV reduction. This is consistent with the unique pasting properties of the potato starch compared to the wheat and chickpea starches. On the previously reported DSC data, this behavior was observed [24]. The maximum PV reduction of the potato starch control was 2829 cP due to GC and by 4073 cP because of GZ. Higher GZ gum concentrations resulted in a greater reduction in the PV of the potato starch control, with the 5% native or acetylated GC or GZ exhibiting the lowest PV compared (Table 4). When compared to the control, the PV of the chickpea starch/GC blend increased dramatically, whereas the GZ blends exhibited a reduction in the PV. The PV of the chickpea starch control increased by 1007 cP and reduced by 790 cP due to GC and GZ, respectively. This reduction in the PV due to GZ is common between the potato and chickpea starches, which is not observed for the wheat starch (Table 4). Given that in some cases greater gum concentrations resulted in a decreased PV when compared to lower concentrations, it is safe to suggest that there is gum-agglomeration at higher concentrations, particularly in the GZ blends with the potato and chickpea starches (Table 2). The decrease in starch PV can be attributed to more than one reason, i.e., the low swelling of the starch granules because of the accumulation of the gum on the surface of the granules or it could be due to the reduction in the amount of the starch—since this is a replacement study (replace starch with gum)—in addition to the nature of the gum (gum makes water unavailable for starch swelling) [24,34]. Unlike potato starch, the final viscosity of the wheat starch and chickpea starch blends was much higher than the PV, despite the fact that the potato starch had the most PV. The data in Table 2 suggests an interaction between GZ and amylose because of the setback of the GZ-starch blend, which was lower than the control (setback is highly dependent on amylose content). Furthermore, because this is a replacement trial, the reduction in setback could be attributable to the lower amylose retrogradation rather than the reduction in starch. For example, when 5% of the starch was substituted with gum, the reduction in setback was significantly greater than 5%, indicating the gum's impact on setback. The wheat starch exhibited the highest pasting temperature, which signifies a delayed start of the pasting process compared to the other starches. Due to the addition of the gum, other researchers noticed an increase in the pasting temperature [35].

**Table 4.** Pasting properties of wheat, potato, and chickpea starch gels containing different levels of *Cordia* and *Ziziphus* gums.

| | [1] PV (cP) | [2] FV (cP) | [3] SB (cP) | [4] PT (°C) |
|---|---|---|---|---|
| | **Wheat starch** | | | |
| | *Cordia* gum | | | |
| 0 | $2970 \pm 18$ [e] | $3725 \pm 38$ [e] | $1384 \pm 25$ [c] | $86.53 \pm 0.09$ [a] |
| 2% native | $3363 \pm 07$ [d] | $4044 \pm 20$ [d] | $1538 \pm 22$ [b] | $64.38 \pm 0.08$ [d] |
| 2% acetylated | $3707 \pm 20$ [c] | $4483 \pm 52$ [a] | $1672 \pm 39$ [a] | $66.45 \pm 0.35$ [b] |
| 5% native | $4172 \pm 16$ [a] | $4465 \pm 0.5$ [b] | $1336 \pm 19$ [c] | $65.99 \pm 0.07$ [c] |
| 5% acetylated | $3876 \pm 04$ [b] | $4240 \pm 0.4$ [c] | $1386 \pm 0.7$ [c] | $66.05 \pm 0.04$ [c] |

**Table 4.** *Cont.*

|  | ¹ PV (cP) | ² FV (cP) | ³ SB (cP) | ⁴ PT (°C) |
|---|---|---|---|---|
| | | **Wheat starch** | | |
| | | *Ziziphus* **gum** | | |
| 0 | 2970 ± 18 ᵈ | 3725 ± 38 ᵇ | 1384 ± 25 ᵃ | 86.53 ± 0.09 ᵇ |
| 2% native | 3025 ± 04 ᶜ | 3634 ± 0.4 ᶜ | 1277 ± 10 ᵇ | 86.50 ± 0.03 ᵇ |
| 2% acetylated | 3271 ± 08 ᵃ | 3964 ± 0.7 ᵃ | 1274 ± 0.7 ᵇ | 84.90 ± 0.04 ᶜ |
| 5% native | 3039 ± 15 ᶜ | 3452 ± 27 ᶜ | 1209 ± 0.9 ᶜ | 88.11 ± 0.02 ᵃ |
| 5% acetylated | 3162 ± 34 ᵇ | 3730 ± 0.4 ᵇ | 1122 ± 23 ᵈ | 84.90 ± 0.04 ᶜ |
| | | **Potato starch** | | |
| | | *Cordia* **gum** | | |
| 0 | 9829 ± 171 ᵃ | 4395 ± 77 ᵃᵇ | 644 ± 14 ᵃᵇ | 64.47 ± 0.02 ᶜ |
| 2% native | 7005 ± 37 ᵈ | 3904 ± 77 ᶜ | 742 ± 48 ᵃ | 66.13 ± 0.04 ᵃ |
| 2% acetylated | 8573 ± 85 ᵇ | 4465 ± 0.5 ᵃ | 402 ± 33 ᶜ | 64.23 ± 0.55 ᵇ ᶜ |
| 5% native | 8136 ± 89 ᶜ | 4352 ± 10 ᵃᵇ | 232 ± 14 ᵈ | 65.19 ± 0.03 ᵇ |
| 5% acetylated | 8175 ± 49 ᶜ | 4305 ± 70 ᵇ | 554 ± 88 ᵇ | 64.83 ± 0.33 ᶜ |
| | | *Ziziphus* **gum** | | |
| 0 | 9829 ± 171 ᵃ | 4395 ± 77 ᵇᶜ | 644 ± 14 ᵃ | 64.47 ± 0.02 ᵇ |
| 2% native | 7673 ± 0.5 ᵇ | 4463 ± 69 ᵃᵇ | 350 ± 50 ᵇ | 64.83 ± 0.36 ᵇ |
| 2% acetylated | 7509 ± 41 ᵇ | 4563 ± 18 ᵃ | 538 ± 70 ᵃ | 64.88 ± 0.38 ᵇ |
| 5% native | 5756 ± 48 ᵈ | 4159 ± 72 ᵈ | 619 ± 71 ᵃ | 66.39 ± 0.31 ᵃ |
| 5% acetylated | 6274 ± 0.3 ᶜ | 4268 ± 70 ᶜᵈ | 659 ± 76 ᵃ | 66.13 ± 0.04 ᵃ |
| | | **Chickpea starch** | | |
| | | *Cordia* **gum** | | |
| 0 | 4331 ± 27 ᶜ | 5822 ± 0.8 ᵈ | 2896 ± 63 ᵈ | 69.83 ± 0.28 ᵃ |
| 2% native | 5991 ± 36 ᵃ | 5592 ± 29 ᵉ | 3777 ± 14 ᶜ | 64.39 ± 0.03 ᵈ |
| 2% acetylated | 5576 ± 64 ᵇ | 6442 ± 54 ᵇ | 5172 ± 32 ᵇ | 63.88 ± 0.31 ᵈ |
| 5% native | 5519 ± 85 ᵇ | 6044 ± 40 ᶜ | 2644 ± 22 ᵉ | 66.48 ± 0.33 ᵇ |
| 5% acetylated | 5338 ± 30 ᶜ | 6873 ± 20 ᵃ | 5298 ± 0.7 ᵃ | 65.22 ± 0.02 ᶜ |
| | | *Ziziphus* **gum** | | |
| 0 | 4331 ± 27 ᵃ | 5822 ± 0.8 ᵃ | 2896 ± 63 ᵃ | 69.83 ± 0.28 ᶜ |
| 2% native | 3927 ± 20 ᶜ | 5160 ±3 7 ᶜ | 2534 ± 29 ᶜ | 70.87 ± 0.02 ᵇ |
| 2% acetylated | 4153 ± 22 ᵇ | 5522 ± 32 ᵇ | 2650 ± 56 ᵇ | 70.93 ± 0.10 ᵇ |
| 5% native | 3541 ± 0.3 ᵉ | 4532 ± 20 ᵈ | 2060 ± 24 ᵈ | 69.36 ± 0.02 ᶜ |
| 5% acetylated | 3607 ± 10 ᵈ | 4550 ± 21 ᵈ | 2007 ± 28 ᵈ | 72.28 ± 0.35 ᵃ |

Mean carrying same letters in columns for particular gum under different starches are non-significantly different from each other. ¹ PV = peak viscosity; ² FV = final viscosity; ³ SB = setback; ⁴ PT = pasting temperature.

### 3.5. Freeze–Thaw Stability of Starch Gels

Syneresis is the release of water from a starch gel following repeated freezing and thawing cycles, which is mostly induced by amylose retrogradation. Figure 2 shows the freeze–thaw stability results, which demonstrates that both gums reduced the syneresis of the wheat and chickpea starches. The reduction was starch concentration dependent, i.e., syneresis was lower at higher gum concentrations. The potato starch showed no syneresis, or very little compared to the other starches, but the addition of gum induced a significant increase in syneresis, but it was much lower than the other starches. Except for the chickpea starch, the syneresis was significantly higher after eight days of storage (Figure 2). The syneresis of the potato starch changed just slightly, with a modest rise in syneresis. After four days, the potato starch control did not release any water, and after eight days, the syneresis remained exceedingly low in comparison to the other two starches. Chickpea also released a lot less water after eight days compared to four days, which indicates limited amylose retrogradation. When compared to the other starches, the wheat starch showed the most syneresis. The wheat starch-GC blend had the highest syneresis and continued to lose water after eight days, indicating that amylose retrogradation was significant. Generally, the reduction in syneresis can be attributed to the ability of the gum to prevent amylose

retrogradation, as reported by other researchers [36]. Gums can promote syneresis in some cases, possibly due to their aggregation and inability to interact with amylose, which was suggested before regarding the PV [24,25].

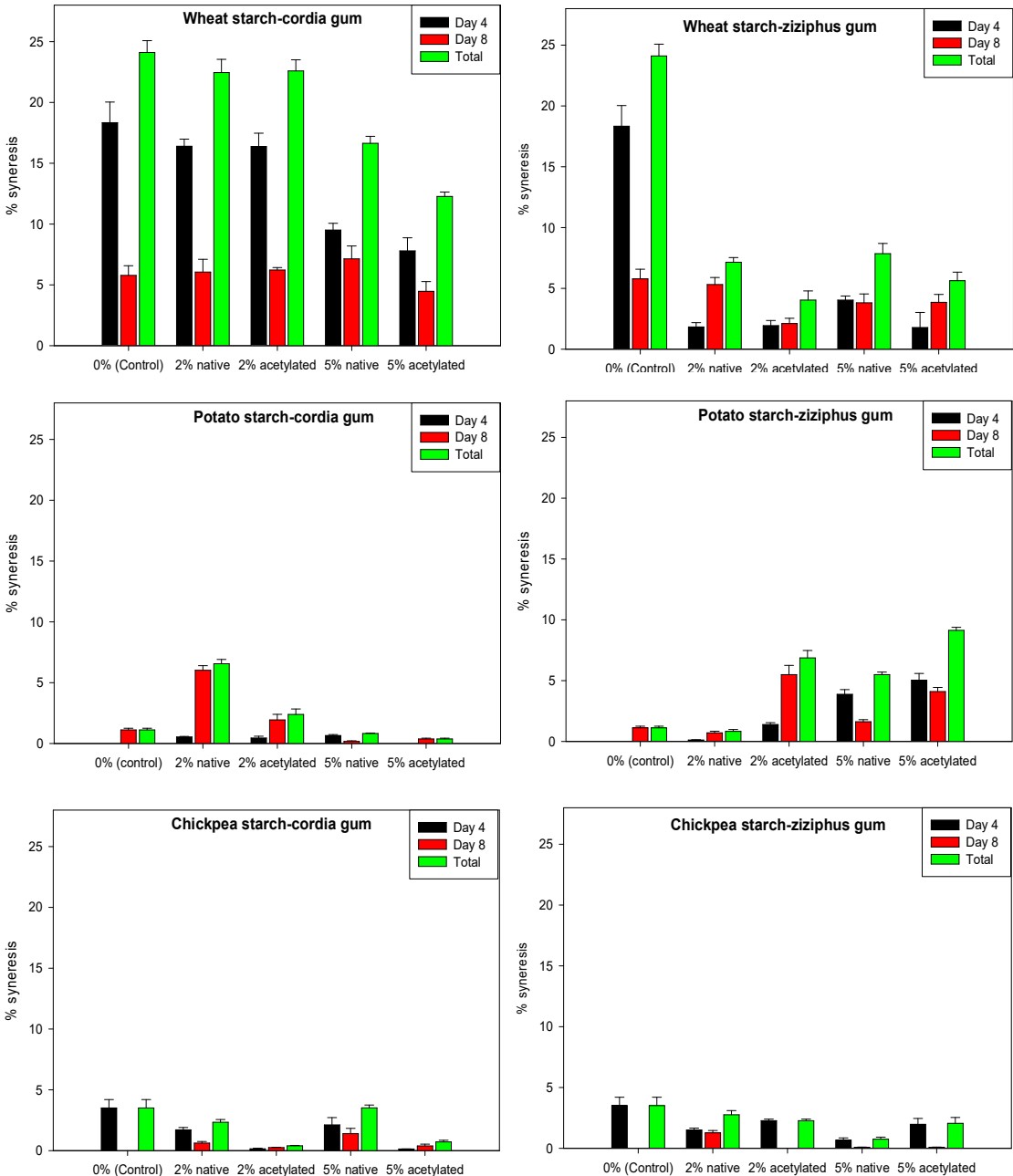

**Figure 2.** % Syeresis from wheat, potato, and chickpea starch gels containing different levels of *Cordia* and *Ziziphus* gums.

### 3.6. Gel Texture

The textural parameters described here include the hardness, which is the force required for sample deformation; cohesiveness, which is the strength of internal bonding in the sample; and gumminess, which is the energy required to breakdown a semi-solid food in the mouth until it is ready to swallow (hardness × cohesiveness). The texture of the gels was evaluated after they had been stored overnight (Figure 3). When compared to the control, the wheat- or chickpea-GC blends had a substantial ($p < 0.05$) increase in hardness, while the acetylated GC blends had the least hardness. The association (retrogradation) of

amylose molecules, a method by which amylose forms a network by hydrogen bonding and traps the water, results in harder gels. The softer gel in the presence of acetylated GC could be due to a lower amylose retrogradation. This was evident in the syneresis data, where low syneresis indicates little amylose retrogradation. The gel hardness of the control chickpea starch (no gum added) was four to seven times that of the wheat and potato, respectively, but the addition of gum increased it even more. The 5% acetylated GZ was the only case where gum addition lowered the hardness of the chickpea starch. The hardness of the potato starch gels differed depending on the gum type, with GC considerably lowering the hardness and GZ greatly increasing it. The data for the potato starch differed from the data for the other starches, as it did for the other tests outlined above. These data support prior research that suggests an increased gum concentration leads to a softer gel [34]. Once again, the gum aggregation could be responsible for variations in the gum effectiveness in terms of the gel hardness (Table 4). Furthermore, the findings revealed a range of cohesiveness, indicating changes in the internal forces of the starch gel with and without gum. Figure 3 shows that GC improved the cohesiveness of the wheat starch gels when compared to the chickpea and potato starch, but the GZ effect varied depending on the starch type and concentration. The behavior of the potato starch in terms of cohesiveness was once more different from that of the other starches because GZ increased the cohesiveness of the potato starch, unlike chickpea, which indicates stronger internal forces in the gel and rules out the idea of gum aggregation. The top three gel hardnesses of the tested starches with GC can be ranked as chickpea–2% native gum > chickpea–5% native gum > chickpea–2% acetylated gum, whereas GZ ranks were: chickpea–2% native gum > chickpea_2% acetylated gum > chickpea_0% native. It's clear that the chickpea starch/gum blends, native or acetylated, exhibited the highest gel hardness, and potato starch the softest. The amylose content and amylose chain length of these starches is the key distinction [37].

*3.7. Rheological Measurements and Temperature Dependency*

Usually, the elastic (G′), viscose (G″) moduli, and complex viscosity ($\eta^{*)}$) are the most discussed parameters in dynamic rheological studies of most food systems. G′ reflects the amount of energy stored and recovered during oscillation (solid property), whereas G″ represents the lost energy (viscous nature). By establishing optimum testing settings (oscillation frequency, strain, and temperature) that are within the linear viscoelastic region (LVR) of the tested gels, the number of deviations from these parameters due to the testing conditions can be ascertained. The LVR of the present work was determined within a wide range of experimental temperatures. Nonetheless, a 5% strain between 25 and 50 °C was found within the LVR. Most of the literature reports used up to 50% strain at a range of 10 to 47 (rad/s) frequencies [38]. However, the 5% strain utilized in this study is low enough to be within the LVR and allows for gel characterization without altering its structure. For most starchy products, a number of researchers suggested the use of G′ as a guide to determine the experimental conditions rather than G″.

Blends with either gum are considered elastic gels rather than viscous, since G′ was substantially greater than G″ (Figures 4 and 5). The starch gels behaved differently in relation to the G′ depending on the starch source or gum type and its concentration. Figure 4 showed that the wheat starch gels with 2% acetylated GC or GZ had the greatest G′, indicating a more elastic gel, but a higher degree of acetylation reduced the G′. This behaviour was observed for the gel hardness (Table 5), where higher acetylation formed a softer gel. The G′ profiles of GC-wheat starch gels were closer together compared to the GZ profiles, indicating less variations between the gels. The potato starch control was more viscous than the blends, but elastic gels were detected for the acetylated GC or GZ. The highest G′ was shown for the 5% native GC and 2% acetylated GZ. The profiles of the potato starch-GC blends were at a distance from one another, indicating a bigger variance in G′ amongst the blends. It's also evident that the GC blends have more elasticity than the GZ blends (Figure 4). The control chickpea starch (no gum added) was the only native starch that had the most G′ compared to the blends. The GZ-chickpea blends exhibited much

more elastic property than the GC blends, which was shown by the profiles in Figure 4. The profiles in Figure 4 reveal that the wheat and potato starch blends with either of the gums were frequency-dependent, whereas the chickpea starch blends were substantially less frequency-dependent and had the greatest G′ of the three starch gels. Therefore, the magnitude of the G′ increased with the increase in frequency except for the chickpea starch. A similar trend was reported for the starch/gum blends [28].

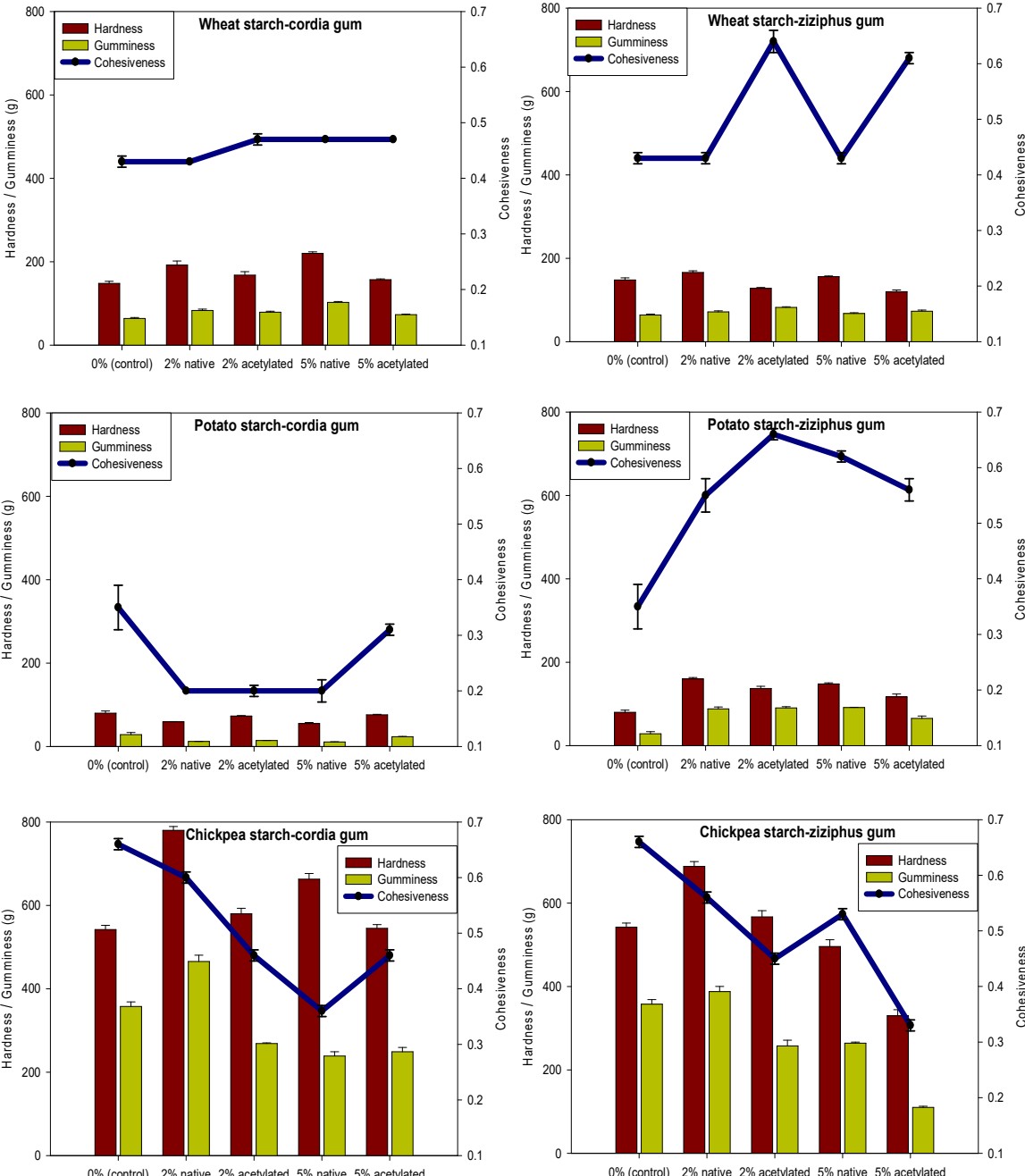

**Figure 3.** Textural properties of wheat, potato and chickpea starch gels containing different levels of *Cordia* and *Ziziphus* gums.

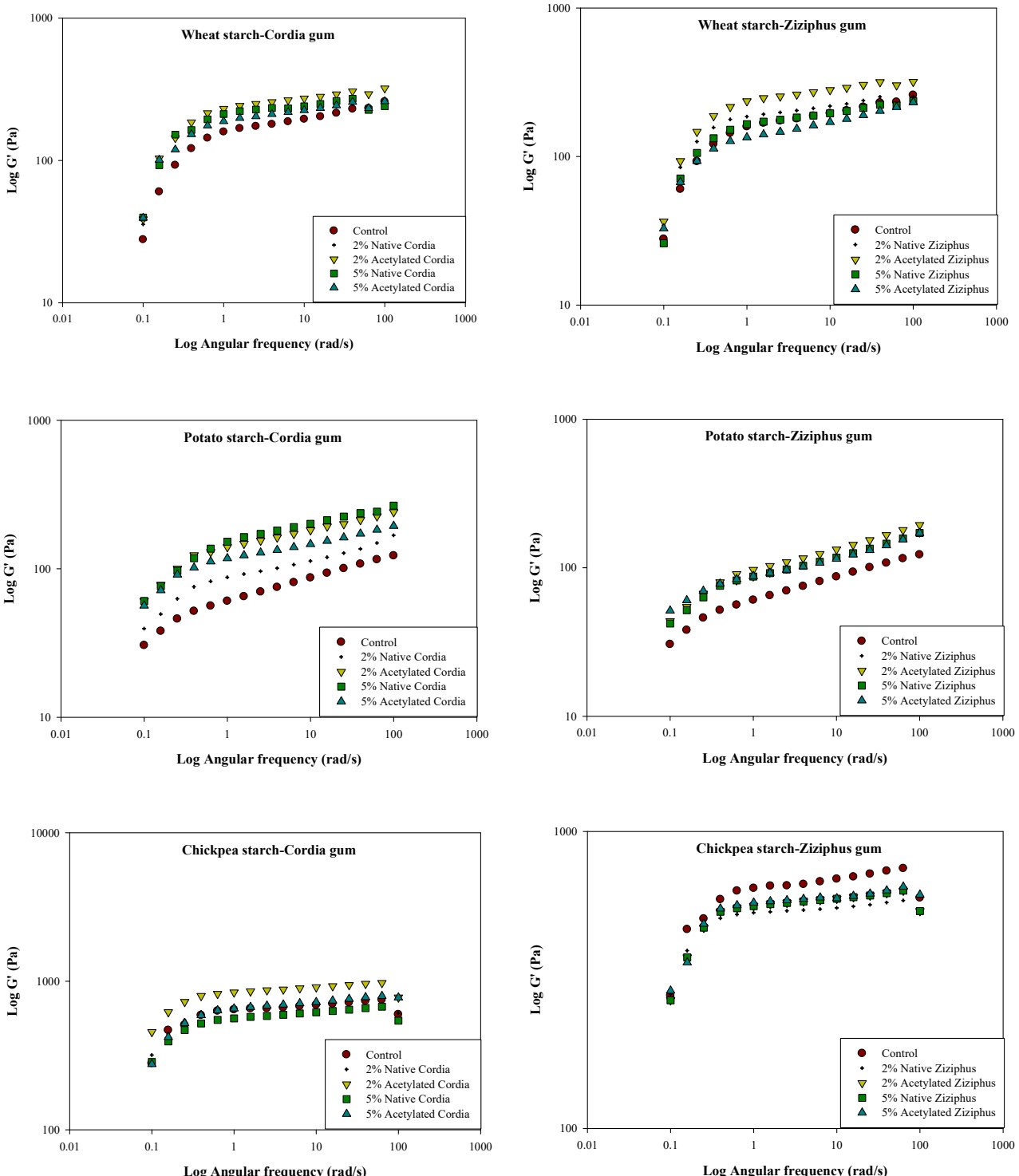

**Figure 4.** Effect of *Cordia* and *Ziziphus* gums on the storage modulus (G′) of different starches.

Certainly, amylose content is the most essential component that caused this discrepancy between the investigated starches, as it regulates the gel's hardness and final structure. Alternatively, we can compare the G′ of the studied starch blends by using 1.0 (rad/s) as a baseline (Table 5). The wheat starch blends with 2% acetylated GC or GZ had the highest G′ at 1.0 (rad/s), whereas the control and the 5% acetylated had the lowest for GC and GZ, respectively. In comparison to the high degrees of acetylation, it is reasonable to infer that low degrees of acetylation allow for the development of the most elastic gel (high G′). Moreover, at 1.0 (rad/s), both gums blended with wheat starch had comparable

G′ magnitudes (Figure 4). The same was true for the potato starch, with the exception of the 5% native. The GC and GZ gums had varying effects on the G′ of the chickpea starch, where the 2% acetylated GC got the highest G′, while the control chickpea starch got the highest G′ compared to the GZ blend (Table 5). Other tested parameters, such as gel texture, showed the different behaviors of the chickpea starch compared to the other starches. Given that the G′ of the native gum mixes is generally low, 2% acetylated gum can be recommended for applications that require elastic gel.

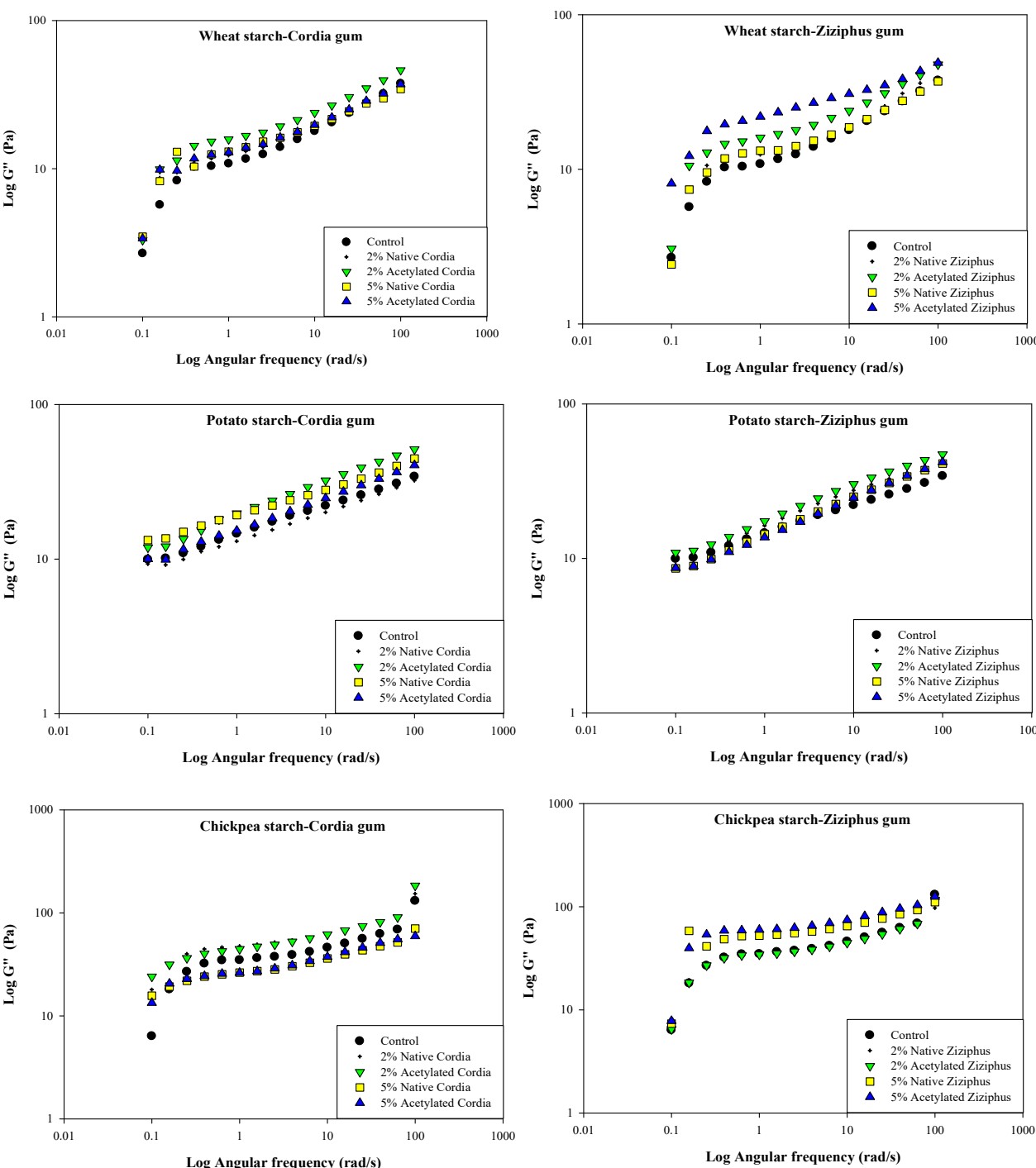

**Figure 5.** Frequency sweep of the Effect of *Cordia* and *Ziziphus* gums on the loss modulus (G″) of different starches.

The shear rate ($\gamma$) and shear stress (t) data for the starch and gum blends at various concentrations and temperatures were well fitted to the power law equation (Equation (1)) and are presented in Figures 6 and 7. The data showed high coefficients of determination ($R^2$), which indicates the appropriateness of the power law for the presented data. All of the examined materials are characterized as pseudoplastic, since the n values were < 1.0. Shear thinning was observed in all samples with *n* values that range between 0.22 and 0.63. The *n* value decreased with an increase in gum concentration. The reduction in the *n* value indicates an inclination towards pseudoplasticity. For the same type of starch, the acetylated gum reduced the *n* value even further than the native gum (Figure 6). The effect of xanthan gum on rice starch was reported to reduce the *n* value as a function of the gum concentration [31]. Therefore, the shear thinning behavior and the power law parameters of the data presented here are in agreement with the literature [39,40]. The change in the *n* value of the specific blend at a higher temperature was limited, whereas the k value decreased with an increase in the temperature from 30 to 50 °C. With the addition of gum, the amplitude of the consistency coefficient (K) indicated by the power law of the wheat and chickpea starches increased, especially at higher gum concentrations, and reached a maximum at 11.4 and 4.98 Pa $s^n$ for GC and GZ, respectively. The chickpea starch, on the other hand, increased in K and reached a maximum of 21.4 and 7.2 Pa $s^n$ for GC and GZ, respectively. The addition of gum lowered the K value of the potato starch, especially at higher concentrations. Unlike the GZ, the starch's K value increased as the GC gum content increased. A higher GC concentration increased the K value of the chickpea starch gel, whereas a higher GZ concentration seemed to have a limited effect on the K value—regardless of whether the gum was native or acetylated (Figure 6). The maximum stress was recorded for gels prepared from the GC blends, especially the acetylated, as shown in the profiles of Figure 7. It's also true that the samples with the highest GC concentration were under the most stress (5%). The rheological properties of the starch/gum blends were presumably dependent on the gum concentration, according to the data presented here.

**Table 5.** The effect of gum *Cordia* and *Ziziphus* on the G′ of wheat, potato, and chickpea starches at 1.0 (rad/s).

| Wheat Starch | |
|---|---|
| **Gum Type** | G′ Rank |
| GC | 2% AC > 5% NA > 2% NA > 5% AC > C |
| GZ | 2% AC > 2% NA > 5% NA > C > 5% AC |
| Potato starch | |
| GC | 5% NA > 2% AC > 5% AC > 2% NA > C |
| GZ | 2% AC > 2% NA > 5% AC > 2% NA > C |
| Chickpea starch | |
| GC | 2% AC > 5% NA > 2% NA > 5% AC > C |
| GZ | C > 5% AC > 2% AC > 5% NA > 2% NA |

GC = gum *Cardia*; GZ = gum *Ziziphus*; AC = acetylated; NA = native; C = control chickpea starch showed decrease with both gums. The decrease in *Ea* indicates less influence of temperature on the parameters, however *Ea* was concentration dependent (Table 6). In general, higher native gum concentration resulted in increased *Ea*, whereas higher acetylated gum instigated lower *Ea*. Therefore, the *Ea* blends varied across blends where the same gum (GC) prompted different effects on the wheat and chickpea starches. The control wheat starch had the greatest *Ea* value, while 5% native GC blended with the chickpea starch had the lowest. Variation in the *Ea* value of starch–gum blends has been investigated in the literature [28,41] with results that are comparable to those presented here. Consequently, both starch and gum types could be responsible for diversity of the *Ea*.

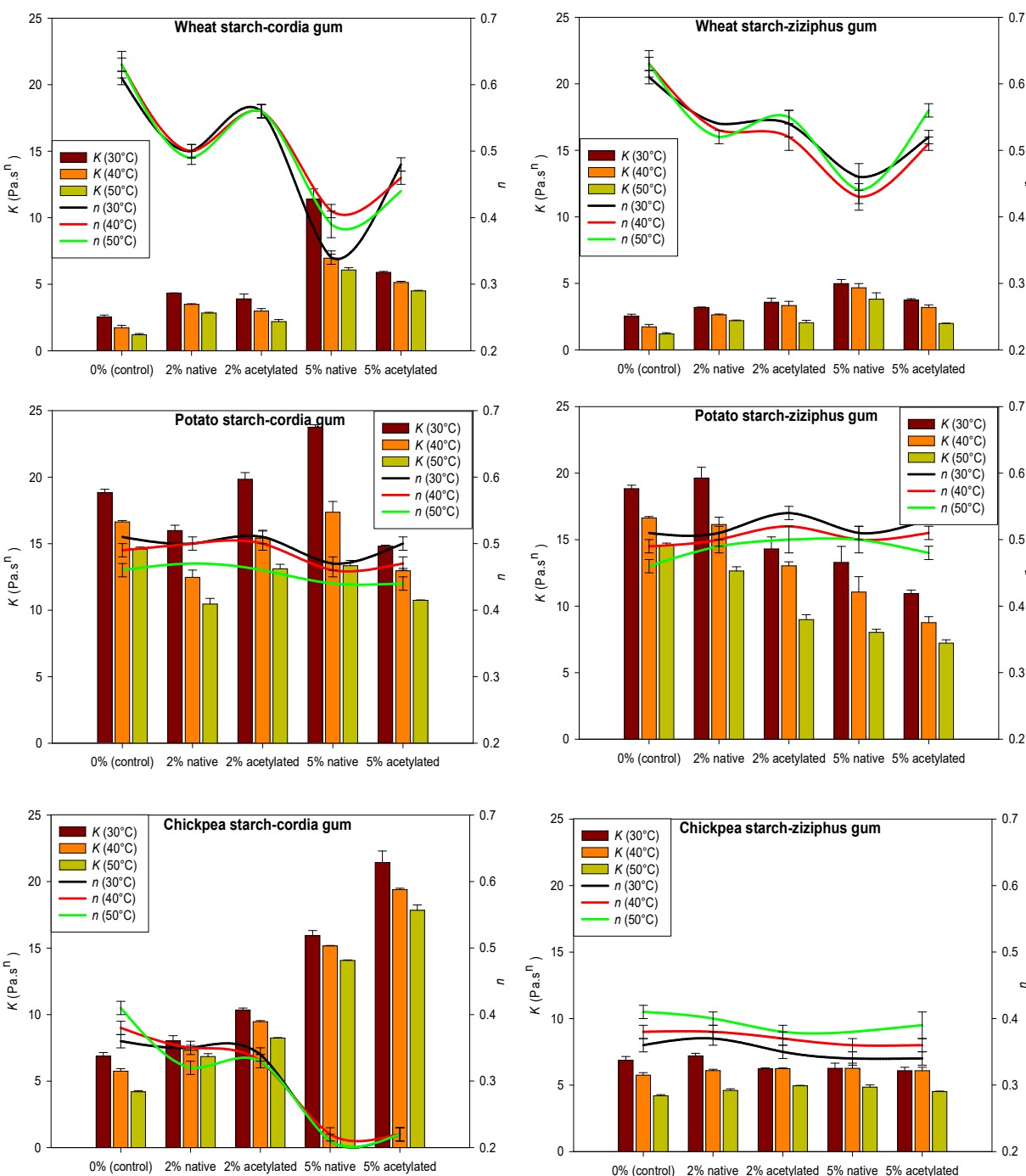

**Figure 6.** Rheological parameters (Power law model) of wheat, potato, and chickpea starch gels containing different levels of *Cordia* and *Ziziphus* gums.

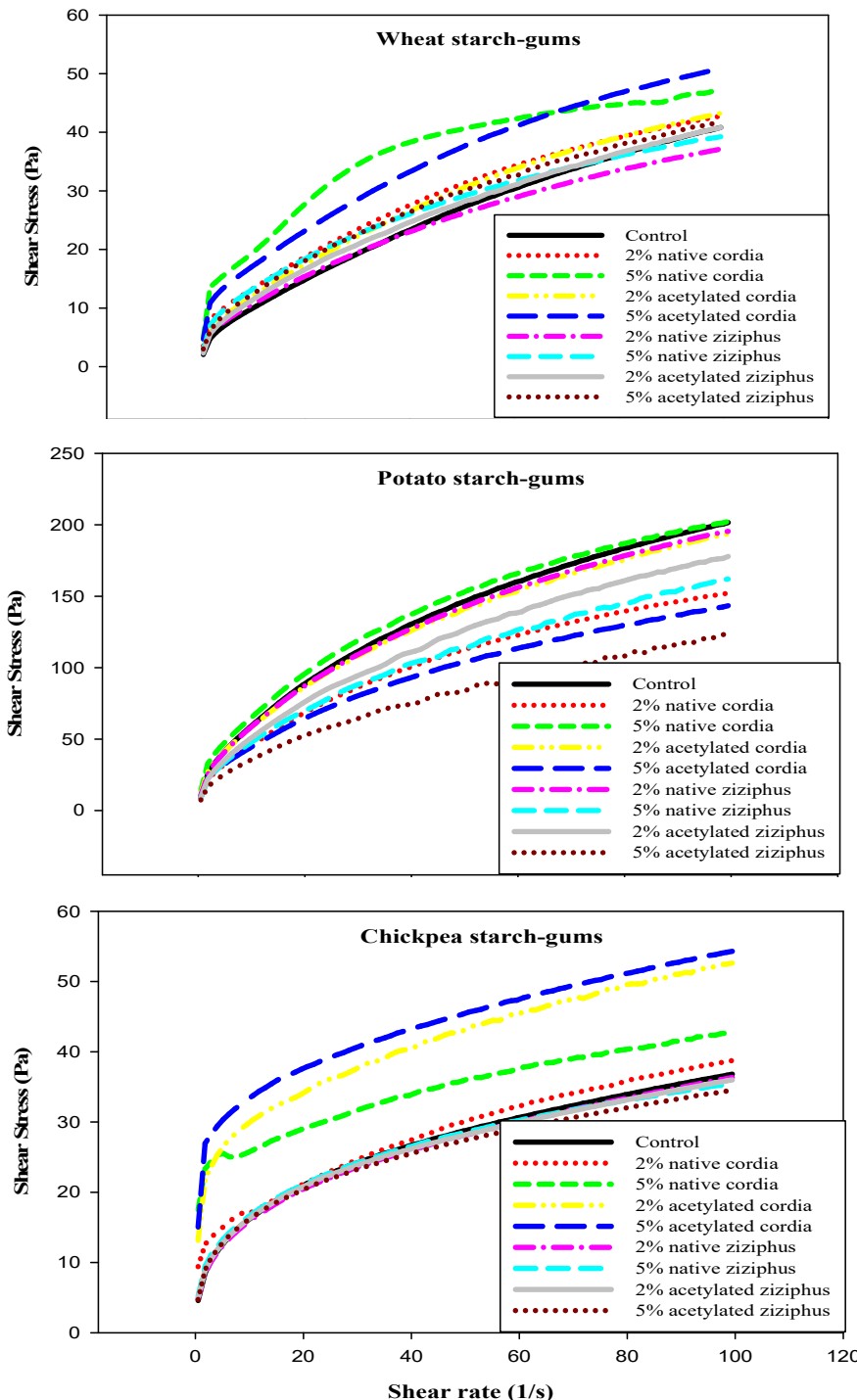

**Figure 7.** Shear stress (Pa) vs. shear rate ($s^{-1}$) plots of wheat, potato, and chickpea starch blends with *Cordia* and *Ziziphus* gums 30 °C.

The Arrhenius equation, which is based on plotting the log of the apparent viscosity (Pa s) at $100 \, s^{-1}$ ($\mu$) against the inverse of the temperature (Kelvin) and determining the activation energy (*Ea*) from the slope of the line, can be used to determine the effect of temperature on the rheological properties of high moisture biomaterials at a given shear rate (Equation (2)). The calculated *Ea* values were in the range of 10,982–30,420 (J/mol $k^{-1}$), and were in the range of 10,866–25,782 for wheat starch-GC and GZ, respectively, with a high determination coefficient (0.92–0.99)—except for the 2% acetylated GZ (Table 6). The

native starches exhibited *Ea* as, wheat starch > chickpea > potato. Overall, the *Ea* of the blends exhibited an increase for the potato starch blends and wheat-GZ blend.

**Table 6.** Activation energy parameters of wheat, potato, and chickpea starches containing *Cordia* and *Ziziphus* gums.

| Starch Blends | Upward Curves | | | Downward Curves | | |
|---|---|---|---|---|---|---|
| | $\mu$ (Pa s$^n$) [a] | $E_a$ (J/mol K$^{-1}$) [b] | $R^2$ | $\mu$ (Pa s$^n$) [a] | $E_a$ (J/mol K$^{-1}$) [b] | $R^2$ |
| **Wheat Starch** | | | | | | |
| *Cordia* gum | | | | | | |
| 0 | $7.26 \times 10^{-12}$ | 30,420 | 0.99 | $2.86 \times 10^{-6}$ | 16,474 | 0.98 |
| 2% native | $4.22 \times 10^{-6}$ | 17,235 | 0.99 | $8.72 \times 10^{-4}$ | 11,434 | 0.97 |
| 2% acetylated | $1.17 \times 10^{-8}$ | 23,437 | 0.92 | $1.01 \times 10^{-4}$ | 13,516 | 0.98 |
| 5% native | $1.78 \times 10^{-8}$ | 25,798 | 0.92 | $1.74 \times 10^{-5}$ | 17,810 | 0.84 |
| 5% acetylated | $2.60 \times 10^{-3}$ | 10,982 | 0.99 | 0.06 | 7616 | 0.94 |
| *Ziziphus* gum | | | | | | |
| 2% native | $1.57 \times 10^{-5}$ | 15,023 | 0.99 | $1.89 \times 10^{-4}$ | 12,460 | 0.99 |
| 2% acetylated | $2.08 \times 10^{-8}$ | 19,430 | 0.84 | $1.29 \times 10^{-3}$ | 10,548 | 0.94 |
| 5% native | $2.07 \times 10^{-3}$ | 10,866 | 0.91 | 0.01 | 9005 | 0.99 |
| 5% acetylated | $1.38 \times 10^{-9}$ | 25,782 | 0.92 | $2.81 \times 10^{-4}$ | 12,430 | 0.98 |
| **Potato starch** | | | | | | |
| *Cordia* gum | | | | | | |
| 0 | 0.06 | 10,426 | 0.99 | 0.01 | 12,533 | 0.99 |
| 2% native | $7.99 \times 10^{-5}$ | 17,285 | 0.99 | $1.06 \times 10^{-3}$ | 14,660 | 0.99 |
| 2% acetylated | $1.87 \times 10^{-4}$ | 16,894 | 0.98 | $8.15 \times 10^{-4}$ | 15,693 | 0.99 |
| 5% native | $6.77 \times 10^{-7}$ | 19,277 | 0.99 | $2.99 \times 10^{-3}$ | 14,556 | 0.99 |
| 5% acetylated | $3.41 \times 10^{-3}$ | 13,036 | 0.98 | 0.02 | 11,468 | 0.98 |
| *Ziziphus* gum | | | | | | |
| 2% native | $7.98 \times 10^{-5}$ | 17,858 | 0.99 | $6.44 \times 10^{-3}$ | 13,341 | 0.99 |
| 2% acetylated | $1.78 \times 10^{-5}$ | 18,790 | 0.88 | $6.70 \times 10^{-4}$ | 15,213 | 0.99 |
| 5% native | $3.12 \times 10^{-6}$ | 20,461 | 0.97 | $1.50 \times 10^{-3}$ | 14,251 | 0.99 |
| 5% acetylated | 1.00 | 16,936 | 0.99 | $6.97 \times 10^{-4}$ | 14,673 | 0.99 |
| **Chickpea starch** | | | | | | |
| *Cordia* Gum | | | | | | |
| 0 | $1.08 \times 10^{-6}$ | 19,945 | 0.97 | $7.99 \times 10^{-8}$ | 10,369 | 0.99 |
| 2% native | 0.33 | 7218 | 0.99 | 0.23 | 7115 | 0.99 |
| 2% acetylated | 0.05 | 9245 | 0.98 | 2.20 | 5168 | 0.96 |
| 5% native | 5.96 | 5038 | 0.98 | 0.94 | 7230 | 0.99 |
| 5% acetylated | 1.27 | 7458 | 0.99 | 22.17 | 4404 | 0.99 |
| *Ziziphus* gum | | | | | | |
| 2% native | $6.08 \times 10^{-6}$ | 18,174 | 0.97 | 0.02 | 9599 | 0.99 |
| 2% acetylated | $5.13 \times 10^{-5}$ | 15,854 | 0.98 | 0.03 | 9141 | 0.99 |
| 5% native | $2.32 \times 10^{-5}$ | 16,744 | 0.97 | 0.02 | 9591 | 0.98 |
| 5% acetylated | $2.96 \times 10^{-6}$ | 18,964 | 0.97 | $5.10 \times 10^{-3}$ | 11,026 | 0.99 |

($\ln\mu = \ln A + Ea/R*1/T$); [a] $\mu$ = is the apparent viscosity (Pa s) at 100 s$^{-1}$ A = frequency factor, R = gas constant, [b] *Ea* = energy of activation, T = absolute temperature (Kelvins).

## 4. Conclusions

Starches isolated from wheat, chickpea, and potato were discriminated in terms of the physicochemical properties of the starch or blends. This was observed in their pasting properties, syneresis, gel hardness, and rheology, as well as their thermal properties. The gums inhibited amylose retrogradation, as evidenced by the reduced setback and syneresis of the gels. The gels made with GC, particularly the acetylated and those with the highest GC content (5%), showed the most shear stress. The GC mix gels exhibited better elasticity than the GZ gels, while the control chickpea starch gel (no gum) had the greatest G′. The

gels prepared with gum *Ziziphus*-chickpea blend were significantly more elastic than the gum *Cardia* blends. The effect of the gum on the starch was described well by the Arrhenius relationship with a high R$^2$. Furthermore, the activation energies of the native starches were higher than the blends. Moreover, *Ea* was gum concentration dependent. As a function of the frequency, the dynamic rheological data (G$'$) at 0, 40, and 50 °C showed that the gels displayed a pseudoplastic behaviour (weak gel). Furthermore, unlike the chickpea and potato starches, the wheat starch's flow behaviour index decreased as the gum concentration increased, indicating that the mix became more pseudoplastic.

**Author Contributions:** Conceptualization, S.H. and A.M.; Methodology, M.S.A. and S.H.; Software, A.A.A.Q. and M.A.I.; Validation, S.H. and A.M.; Formal Analysis, M.A.I. and A.M.; Investigation, S.H. and M.S.A.; Resources, M.A.I. and S.H.; Data Curation, A.M. and I.A.A.; Writing—Original Draft Preparation A.M., A.A.A.Q. and S.H.; Writing—Review & Editing, I.A.A. and M.A.I.; Visualization, A.A.A.Q. and I.A.A.; Supervision, A.M., S.H. and M.S.A.; Project Administration, M.A.I. and A.A.A.Q.; Funding Acquisition, A.M., S.H. and M.S.A. All authors have read and agreed to the published version of the manuscript.

**Funding:** This Project was funded by the National Plan for Science, Technology and Innovation (MAARIFAH), King Abdulaziz City for Science and Technology, Kingdom of Saudi Arabia, Award Number (14-AGR2277-02).

**Conflicts of Interest:** The authors declare no conflict of interest.

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
