# Peer review of "Physicochemical Properties of Starch Binary Mixtures with Cordia and Ziziphus Gums"

_processes, doi:10.3390/pr10020180_

Round 1
Reviewer 1 Report
The manuscript “Physicochemical Properties of Starch Composites with Cordia and Ziziphus Gums” is about the texture and thermal rheology of the mixtures of two different gums with starches including wheat, potato, and chickpea starches. The authors used RVA, DSC, texture analyzer, and dynamic rheometer to examine these starch gels. They found the influence of the acetylation gums on the elasticity of the starch gels.
It seems to me that both the English qualification and scientific background of the manuscript are too low to be accepted. The author made so many grammatical and scientific mistakes such as word order (for example gum Cordia and gum ziziphus for Cordia gum and ziziphus gum), parallel phrase (line 11-12), punctuation usage, the symbol for the components of rheology (G` for G', G`` for G''), etc.… The organization of the manuscript is very bad for the readers. The authors try to stuff figures 2, 3, and 4 into the paragraph for FTIR. They also hide tabulated data after the conclusion. That brings so much difficulty for readers to follow. Many parameters in table 2 are not analyzed.
The authors do not know what to analyze and compare data in table 2. The authors do not compare the peak temperature obtained by DSC and RVA. So many paragraphs are very poor and easily cause misunderstanding because of the lack of scientific knowledge and poor English writing. For example, the paragraph for FTIR shows the shortage of knowledge in FTIR interpretation for polysaccharide compounds, the paragraphs for DSC, RVA, and rheological measurements are clumsy.
Below are some examples:
Abstract
Please rewrite the abstract for more attractive, correct all grammatical and scientific mistakes.
Introduction
Please rewrite paragraphs 2, 3, and 4 (lines 38-83) for better focusing on the topic.
Materials and methods
Please correct the company name of Sigma Aldrich, the centrifugal speed, and the formulae for calculating the percentage of acetyl and the degree of substitution. Please add a brief explanation for those formulae.
The method on rheological measurement and the results were not consistent. Line 167-168: “frequency sweeps ranging from 0.63-62.8 rad/s at 2% strain”, while line 397-404 “5% strain used in this study is low enough to be within the LVR”.
Results
Lines 177-178: The data from the selected citation does not confirm the information found in the manuscript. For example, the band at 2922 cm-1 represents both the monosaccharide sugars including rhamnose, galactose, and arabinose, and the C-H stretch in aldehyde or alkane groups.
Line 183: The claim “The degree of acetylation was 0.78 and 0.17 for the GC and GZ, respectively” does not make sense and seems to be subjective. FTIR data shows the decrease of the band at 1585-1559 cm-1 for the acetylated samples. However, how the change correlates to the claim is unclear. Furthermore, there is not any new band that appears only in the FTIR spectrum for the acetylated samples to confirm the formation of acetylated products.
Line 268-270 the claim “The OT and PT … patterns” is unclear and can cause misunderstanding. For example, table 1 shows that OT and PT for wheat samples of 5% native gum ziziphus are the same as those of samples with 2% acetylated gum ziziphus. Similarly, the OT and PT of the wheat sample with 2% native gum ziziphus are the same as those of the wheat sample with 5% acetylated gum ziziphus.
Line 274-275 the sentence” Literature reports …temperatures” is self-conflict and incorrect. How the increasing of melting enthalpy of acetylated gums causes the decrease in the gelatinization enthalpy in the wheat samples with 2% and 5% acetylated GC gums. Furthermore, that statement does not relate to the fact in the claim “The gelatinization enthalpy …GZ reduced it at both concentrations” in line 272-273.
Line 278-279 please rewrite this sentence. The cereal starches are naturally different from the tuber starches. This claim indicates the authors do not understand the influence of chemical components on the physical properties of the samples.
Please correct the unit abbreviation for centipoise in the paragraph for rapid visco analyzer.
Line 303-305 the claim “acetylated GC… native GC.” is incorrect. The peak viscosity for samples pure wheat, wheat blend with 2% native GC, and wheat blend with 5% native GC is 2970, 3363, and 4172 cP, respectively. The peak viscosity for the samples of wheat blend with 2% and 5% acetylated GC is 3707 and 3876 cP respectively.
Line 307-309 the claim “acetylated GZ… native GC … concentration dependency.” is also incorrect. The peak viscosity for wheat samples with 2% and 5% native GZ is 3025 and 3039 cP, respectively. The peak viscosity of the wheat samples with 2% and 5 acetylated GZ is 3271 and 3162 cP, respectively.
Line 310-313 the claim “The 5% acetylated … is desired” is incorrect.
Author Response
Abstract
Please rewrite the abstract for more attractive, correct all grammatical and scientific mistakes.
-Was done
Introduction
Please rewrite paragraphs 2, 3, and 4 (lines 38-83) for better focusing on the topic.
-Paragraph was re-written
Materials and methods
Please correct the company name of Sigma Aldrich, the centrifugal speed, and the formulae for calculating the percentage of acetyl and the degree of substitution. Please add a brief explanation for those formulae.
The method on rheological measurement and the results were not consistent. Line 167-168: “frequency sweeps ranging from 0.63-62.8 rad/s at 2% strain”, while line 397-404 “5% strain used in this study is low enough to be within the LVR”.
-Corrected
Results
Lines 177-178: The data from the selected citation does not confirm the information found in the manuscript. For example, the band at 2922 cm-1 represents both the monosaccharide sugars including rhamnose, galactose, and arabinose, and the C-H stretch in aldehyde or alkane groups.
-A subtitle was added to the FTIR section
Line 183: The claim “The degree of acetylation was 0.78 and 0.17 for the GC and GZ, respectively” does not make sense and seems to be subjective. FTIR data shows the decrease of the band at 1585-1559 cm-1 for the acetylated samples. However, how the change correlates to the claim is unclear. Furthermore, there is not any new band that appears only in the FTIR spectrum for the acetylated samples to confirm the formation of acetylated products.
-Was done
Line 268-270 the claim “The OT and PT … patterns” is unclear and can cause misunderstanding. For example, table 1 shows that OT and PT for wheat samples of 5% native gum ziziphus are the same as those of samples with 2% acetylated gum ziziphus. Similarly, the OT and PT of the wheat sample with 2% native gum ziziphus are the same as those of the wheat sample with 5% acetylated gum ziziphus.
-Was corrected
Line 274-275 the sentence” Literature reports …temperatures” is self-conflict and incorrect. How the increasing of melting enthalpy of acetylated gums causes the decrease in the gelatinization enthalpy in the wheat samples with 2% and 5% acetylated GC gums. Furthermore, that statement does not relate to the fact in the claim “The gelatinization enthalpy …GZ reduced it at both concentrations” in line 272-273.
-Yes, it was done
Line 278-279 please rewrite this sentence. The cereal starches are naturally different from the tuber starches. This claim indicates the authors do not understand the influence of chemical components on the physical properties of the samples.
-Sentence was rewritten
Please correct the unit abbreviation for centipoise in the paragraph for rapid visco analyzer.
-Done
Line 303-305 the claim “acetylated GC… native GC.” is incorrect. The peak viscosity for samples pure wheat, wheat blend with 2% native GC, and wheat blend with 5% native GC is 2970, 3363, and 4172 cP, respectively. The peak viscosity for the samples of wheat blend with 2% and 5% acetylated GC is 3707 and 3876 cP respectively.
-Corrected
Line 307-309 the claim “acetylated GZ… native GC … concentration dependency.” is also incorrect. The peak viscosity for wheat samples with 2% and 5% native GZ is 3025 and 3039 cP, respectively. The peak viscosity of the wheat samples with 2% and 5 acetylated GZ is 3271 and 3162 cP, respectively.
-The data was corrected
Line 310-313 the claim “The 5% acetylated … is desired” is incorrect.
-the sentence was revised
Reviewer 2 Report
- Ln 18: Key words don’t match the text and abstract. For example: it has not been pointed out in the abstract that is sweet potato! Please modify all the keywords.
- It seems that there is a “double space” between the “full stops” and starting the next sentence in the whole text. Please check them all again.
- Ln 21: The definition of gums needs to be revised. Gums are a sub-category of hydrocolloids and they do not present the same meaning! Ln22, 23 and 24 show a more accurate definition.
- Ln 23: instead “with high molecular weight molecules”: “with high molecular weights”.
- Ln 25, 26 and 27: are not coherent. Please rewrite the sentences and check the grammar points.
- Ln 36, 37 and 38: are not comprehensible. Please rewrite the sentences and check the grammar points.
- Ln 41: The consistency of the text has been suddenly lost. The writer was making examples of the most well-known gums, and then suddenly started introducing Cordia. Please add a short intro sentence, in order to establish a connection between sentences and topics.
- Ln 43 and 44: It is suggested to add a colour photo of Cordia Myxa
- Ln 45 - 50: The text is not understandable. It would be better to summarize all the information of theses sentences in a table.
- Ln 51 and 52: Do you have any idea how much (or %) alkaloids are in the fruit?
- Ln 54 and 55: It will be more useful, if writers add a molecular structure of the gum.
- Ln 61: Edit the reference.
- Ln 57 - 66: It is not understandable why the writers, in the middle of explanation about Cordia gum, have suddenly started introducing Arabic gum!
- Ln 67 and 77- 80: Please grammatically check.
- Ln 81: Add a full stop at the end of the sentence.
- There is no explanation about ziziphus gum in the introduction part! A paragraph should be added to elaborate the source of ziziphus, as well as molecular structure and application of ziziphus gum.
- Why have you chosen chickpea, wheat and potato starch?
- Ln 85: Please grammatically check.
- Subtitles should be also numbered: 2.1. Material/ 2.2. chickpea starch isolation, etc.
- Ln 88: why did you add an address in US in the middle of the sentence? You just need to name the model of the blender.
- Ln 89: Please add the information about the Centrifuge: name of the company, model, etc.
- Ln 87 and 91: what was the temperature of the water?
- What was the reference of the extraction method?
- Ln 97: Please add the information about the Centrifuge: name of the company, model, etc.
- Ln 99: what was the temperature of the water?
- Ln 102: Please mention the place that fruits were collected.
- Ln 104: It is again the address of a company, instead the model and brand of the mixer.
- Ln 105: which temperature?
- Ln 106: how was it neutralized?
- Ln 106: information and condition of freeze dryer?
- Ln 108 and 109: results should not be mentioned in the material and method section.
- Ln 125: the equation should be typed, not copied. Please add the reference.
- Ln 128 - 136: the order of the text, design, titles, full stops, etc. should be checked and revised.
- Ln 133: information about the FTIR is not complete.
- Ln 139 and 140: grammatically check.
- Ln 141: h not hr
- There must be one space between °C and the previous word or number.
- Line 141: order of the line needs to be checked.
- Ln 143: To or Tm, o and m should be added as subscripts
- Ln 144: details of the software?
- Add a paragraph about preparation of controls
- Ln 145: what has been exactly measured here? What was the details of measurement?
- Ln 156 and 157: grammatically check
- Ln 152: how it was calculated? Was a quantitative test or just a qualitative experiment?
- Ln 154: what was the temperature of the centrifuge?
- Ln 160 and 161: font size check
- Ln 161: sec? it is s
- Ln 165: why did you choose 0.628-0.63 rad/s?
- details of the rheological test are not complete
- Ln 167-169: grammatically check.
- Ln 173: edit the numbering of subtitles
- Results of degree of acetylation should be discussed in a separated subcategory.
- Check the format of citations in the text.
- Ln 179-180: Grammar check.
- Figure 1: title of axes is missed
- Figures 2-4: should be accompanied with their explanations in the text, not in 3 pages earlier!
- Discussion of the results should be improved.
- Ln 260 and 261: grammar check.
- Ln 262: avoid using JUST abbreviations in the title.
- Table 1 should be inserted where it is mentioned for the first time.
- With respect to the authors, there are some sentences which are exactly copied from other literatures. The use of words and consistency of the text are suddenly changing. You are not allowed to copy the exact sentences from other sources. I strongly suggest you to rewrite all the copied sentences in the discussion part.
- Relocate table 2.
- The language of Result parts should be fundamentally edited and improved.
- Relocate table 3.
- Relocate table 4.
- Ln 398: LVR or LVE?
- Ln 440: Pas or Pa? double check all the units.
- Conclusion part should be improved.
Author Response
Comments and Suggestions for Authors
- Ln 18: Key words don’t match the text and abstract. For example: it has not been pointed out in the abstract that is sweet potato! Please modify all the keywords.
- It seems that there is a “double space” between the “full stops” and starting the next sentence in the whole text. Please check them all again.
- Done, throughout
- Ln 21: The definition of gums needs to be revised. Gums are a sub-category of hydrocolloids and they do not present the same meaning! Ln22, 23 and 24 show a more accurate definition.
- done
- Ln 23: instead “with high molecular weight molecules”: “with high molecular weights”.
- Was addressed
- Ln 25, 26 and 27: are not coherent. Please rewrite the sentences and check the grammar points.
- done
- Ln 36, 37 and 38: are not comprehensible. Please rewrite the sentences and check the grammar points.
- Gums sources include, plant exudate (gum Arabic), plant structure (pectin, cellulose), seeds (guar gum, locust bean gum), tuber (konjac mannan), algal (agar, carrageenan, alginate) and microbial based such as xanthan gum, curdlan, gellan
- Ln 41: The consistency of the text has been suddenly lost. The writer was making examples of the most well-known gums, and then suddenly started introducing Cordia. Please add a short intro sentence, in order to establish a connection between sentences and topics.
- One of the relatively new sources of plant based gum, is Cordia myxa (lasura), which is a tree grown in tropical and sub-tropical environment. This tree produces fruits with a substantial amount of gum that can potentially be a plant-based gum.
- Ln 43 and 44: It is suggested to add a colour photo of Cordia Myxa
- A coloured picture was added
- Ln 45 - 50: The text is not understandable. It would be better to summarize all the information of theses sentences in a table.
- A table was added
- Ln 51 and 52: Do you have any idea how much (or %) alkaloids are in the fruit?
- I found No useful information on line
- Ln 54 and 55: It will be more useful, if writers add a molecular structure of the gum.
- Unable to find one
- Ln 61: Edit the reference.
- References were edited
- Ln 57 - 66: It is not understandable why the writers, in the middle of explanation about Cordia gum, have suddenly started introducing Arabic gum!
- It was just an example of how gums are found in nature
- Ln 67 and 77- 80: Please grammatically check.
- The present work objectives were to isolate polysaccharides from ziziphus and Cordia myxa fruits and determine whether these isolated polysaccharides have potential to effect the physicochemical properties of starch from different sources (cereal, legume or tuber) which will directly affect the formulation of starchy products
- Chickpea (CP) whole meal was sluried in distilled water (50:50) and blended by a heavy duty blender for 5 minutes (St 553 Benson Rd, San Antonio, TX, US), filtered through 200 mesh sieve and the filtrate was centrifuged at x 2000 g for 15 minutes.
- Ln 81: Add a full stop at the end of the sentence.
- Was done
- There is no explanation about ziziphus gum in the introduction part! A paragraph should be added to elaborate the source of ziziphus, as well as molecular structure and application of ziziphus gum.
- This paragraph was added to the introduction “Zizyphus spina-christi, is a tree species belonging to the botanical family Rhamnaceae native to a vast area of Africa stretching from Mauritania through the Sahara and coasts zones of West Africa to the Red Sea. It is drought resistant, very resistant to heat and can be found in desert areas where groundwater is available. Reports in the literature indicated that the ethanol extract of Ziziphus fruits exhibited flowing properties similar to xanthan gum and higher than guar gum (Thanatcha and Pranee, 2011). Optimum conditions for Ziziphus mucilage extraction were reported as 1:7 water, 60⁰ ˚C and 1:3 ethanol for precipitation. Water holding capacity, oil absorption and emulsion capacity were 73.35 g water/g dry basis, 4.97 g oil/g dry sample and 52.22 % respectively. This data showed that oil absorption higher than guar gum and xanthan gum, but emulsion capacity was lower (Rangklang and Anprung, 2009). The fruits of most Ziziphus cultivars are found to different in shape, size, color, specific gravity, and seed weight (Obeed et al., 2008). A total of five types of phenolic compounds were detected in Napek extract, a Ziziphus type, but only two were identified by TLC as caffeic acid and p- coumaric acid (Muchuweti et al., 2005).”
- Why have you chosen chickpea, wheat and potato starch?
- Because they represent a divers source of starches with different physicochemical properties i.e cereal, legume and tuber, respectively
- Ln 85: Please grammatically check.
- done
- Subtitles should be also numbered: 2.1. Material/ 2.2. chickpea starch isolation, etc
- done .
- Ln 88: why did you add an address in US in the middle of the sentence? You just need to name the model of the blender.
- done
- Ln 89: Please add the information about the Centrifuge: name of the company, model, etc.
- Fisherbrand™ Refrigerated Centrifuge GT2
- Ln 87 and 91: what was the temperature of the water?
- The temperature of the water was 25⁰C.
- What was the reference of the extraction method?
- Reference number [20]
- Ln 97: Please add the information about the Centrifuge: name of the company, model, etc. Fisherbrand™ Refrigerated Centrifuge GT2
- Ln 99: what was the temperature of the water?
- 25⁰C
- Ln 102: Please mention the place that fruits were collected.
- a local farm in Riyadh, Saudi Arabia
- Ln 104: It is again the address of a company, instead the model and brand of the mixer
- done.
- Ln 105: which temperature?
- 25⁰C
- Ln 106: how was it neutralized?
- using 0.50 N HCl
- Ln 106: information and condition of freeze dryer?
- (Alpha 1-4, LD plus, at 0.005 mBarr and -50 ⁰C)
- Ln 108 and 109: results should not be mentioned in the material and method section.
- OK
- Ln 125: the equation should be typed, not copied. Please add the reference.
- done
- Ln 128 - 136: the order of the text, design, titles, full stops, etc. should be checked and revised.
- done
- Ln 133: information about the FTIR is not complete.
- ALPHA ATR, Brukner, Germany.
- Ln 139 and 140: grammatically check.
- The tested parameters of the thermal analysis were enthalpy ∆H (J/g), onset temperature (To), and peak temperature (Tm) using Universal Analysis Software provided by the DSC manufacturer.
- Ln 141: h not hr
- Done
- There must be one space between °C and the previous word or number
- done.
- Line 141: order of the line needs to be checked.
- done
- Ln 143: To or Tm, o and m should be added as subscripts
- done
- Ln 144: details of the software?
- TA instruments
- Add a paragraph about preparation of controls
- The control was considered starch only without gum
- Ln 145: what has been exactly measured here? What was the details of measurement?
- The profile of the tested samples includes the peak viscosity of the formed gel, final viscosity, setback, and pasting temperature.
- Ln 156 and 157: grammatically check
- The separation of water from gels after 4 or 8 days of storage will be reported as %Syneresis
- Ln 152: how it was calculated? Was a quantitative test or just a qualitative experiment?
- Syneresis was calculated by subtracting the separated water after each cycle from the weight before the cycle.
- Ln 154: what was the temperature of the centrifuge?
- Centrifuge temperature was 10 ⁰C
- Ln 160 and 161: font size check
- done
- Ln 161: sec? it is s
- done
- Ln 165: why did you choose 0.628-0.63 rad/s?
- We used 0.1-100 rad/s
details of the rheological test are not complete
The following paragraph was added “Dynamic rheology measurements at frequency sweeps ranging from 0.1-100 rad/s at 2% strain were carried out on RVA-cooked starch samples using DHR- Hybrid Rheometer (TA Instruments, New Castel PA). Experimental data included storage modulus (G´), loss modulus (G´´) and dynamic mechanical loss tangent (tanδ = G′′/G′). Steady shear behaviors (shear rate vs shear stress) and temperature dependency (30, 40 and 50°C) of cooked starch gels were recorded at variable shear rate of 1 to 100/s. The data was fitted to power law model
|
|
where = shear stress (Pa·s), = consistency coefficient (Pa·s), = shear rate (s−1) and = flow behavior index (dimensionless). , while temperature dependency was estimated by using the Arrhenius Equation 3 [23].
- Ln 167-169: grammatically check
- done
- Ln 173: edit the numbering of subtitles
- done
- Results of degree of acetylation should be discussed in a separated subcategory.
- 2 Degree of acetylation
- The degree of acetylation was 0.78 and 0.17 for the GC and GZ, respectively. The substitution of the hydroxyl groups of GC was around 5 time more than that of GZ. This makes acetylated GC much more hydrophobic than GZ and can be utilized in oil based products. The hydroxyl groups of GC appeared to readily available for replacement with acetyl groups that GZ.
- Check the format of citations in the text.
- done
- Ln 179-180: Grammar check.
- Done
- Figure 1: title of axes is missed
- Axis title was added
- Figures 2-4: should be accompanied with their explanations in the text, not in 3 pages earlier!
- Done
- Discussion of the results should be improved.
- Done
- Ln 260 and 261: grammar check.
- Done
- Ln 262: avoid using JUST abbreviations in the title.
- Done
- Table 1 should be inserted where it is mentioned for the first time.
- Done
- With respect to the authors, there are some sentences which are exactly copied from other literatures. The use of words and consistency of the text are suddenly changing. You are not allowed to copy the exact sentences from other sources. I strongly suggest you to rewrite all the copied sentences in the discussion part.
- Reviewed and changed
- Relocate table 2.
- Done
- The language of Result parts should be fundamentally edited and improved.
- Relocate table 3.
- done
- Relocate table 4.
- done
- Ln 398: LVR or LVE?
- It was corrected
- Ln 440: Pas or Pa? double check all the units.
- Corrected
- Conclusion part should be improved.
- Done
Reviewer 2
Comments and Suggestions for Authors
- Page 3: Acetyl (%) formula. Check the letter format, it seems different of the rest of the manuscript on the pdf file
The equation was changed
- Page 7: The legend of the graph covers part of the image. Please make sure the graph is fully visible.
done
- Information on how the data was statistically analyzed and / or the number of times each experiment was carried out is not shown in the manuscript.
Statistical Analysis
Measurements were done in triplicate and the data were analyzed using ANOVA. A factorial design was applied to test for the effects of GC and GZ on starch. Duncan’s multiple range test was applied to compare means at p ≤ 0.05 using the PASW® Statistics 18 software (SPSS Inc., Hong Kong, China P.R.).
Reviewer 3 Report
- Page 3: Acetyl (%) formula. Check the letter format, it seems different of the rest of the manuscript on the pdf file
- Page 7: The legend of the graph covers part of the image. Please make sure the graph is fully visible.
- Information on how the data was statistically analyzed and / or the number of times each experiment was carried out is not shown in the manuscript.
Author Response
Comments and Suggestions for Authors
- Page 3: Acetyl (%) formula. Check the letter format, it seems different of the rest of the manuscript on the pdf file
The equation was changed
- Page 7: The legend of the graph covers part of the image. Please make sure the graph is fully visible.
done
- Information on how the data was statistically analyzed and / or the number of times each experiment was carried out is not shown in the manuscript.
Statistical Analysis
Measurements were done in triplicate and the data were analyzed using ANOVA. A factorial design was applied to test for the effects of GC and GZ on starch. Duncan’s multiple range test was applied to compare means at p ≤ 0.05 using the PASW® Statistics 18 software (SPSS Inc., Hong Kong, China P.R.).
Reviewer 4 Report
In this paper physicochemical properties of starch composites with cordia and ziziphus gums were investigated. The presented work for evaluation is very interesting. The scope of work is extensive.
However, I did notice at work deficiencies which, in my opinion, are necessary to complete in order to make the manuscript understandable by the reader.
Line 194: “…held at 50 °C for 30 seconds and for 4.40 minutes at 95 °C (at 10.23°C/ min) then held at 95 °C for 4 minutes.” Why was the sample held for two different times at 95 ᵒC? I think that it is a mistake. What was the rate of temperature rise from 50 to 95 ᵒC? Please correct and rewrite the sentence.
Line 220: I recommend using an abbreviation ? for shear stress instead of T. It is widely used. Fig. 1 also includes the T parameter, which causes some confusion in the manuscript.
In addition, the shear stress should be expressed in other units.
Line 220: In my opinion, K should be represented in the following unit: Pa·sn.
Line 475: “..The shear rate (T) and shear stress (γ)…” Incorrect marking, please correct it.
Line 486: What is “…k value …”
Table 9: μ (Pa sn)? Please explain what this symbol means? Is the unit surely wrote correctly?
Fig. 2 has low resolution.
Is there any possible explanation what kind of interaction occured between starches and the hydrocolloids used by the authors?
Author Response
Reviewer 4: Comments and Suggestions for Authors
In this paper physicochemical properties of starch composites with cordia and ziziphus gums were investigated. The presented work for evaluation is very interesting. The scope of work is extensive.
However, I did notice at work deficiencies which, in my opinion, are necessary to complete in order to make the manuscript understandable by the reader.
Line 194: “…held at 50 °C for 30 seconds and for 4.40 minutes at 95 °C (at 10.23°C/ min) then held at 95 °C for 4 minutes.” Why was the sample held for two different times at 95 ᵒC? I think that it is a mistake. What was the rate of temperature rise from 50 to 95 ᵒC? Please correct and rewrite the sentence.
The sentence was reworded
Line 220: I recommend using an abbreviation ? for shear stress instead of T. It is widely used. Fig. 1 also includes the T parameter, which causes some confusion in the manuscript.
Was done.
In addition, the shear stress should be expressed in other units.
Yes it was changed to dyne/cm2
Line 220: In my opinion, K should be represented in the following unit: Pa·sn
done
Line 475: “..The shear rate (T) and shear stress (γ)…” Incorrect marking, please correct it.
Corrected
Line 486: What is “…k value …”
The correct symbol is µ = is the apparent viscosity (Pa s) at 100 s-1 , NOT K
Table 9: μ (Pa sn)? Please explain what this symbol means? Is the unit surely wrote correctly?
µ = is the apparent viscosity (Pa s) at 100 s-1
Fig. 2 has low resolution.
Improved
Is there any possible explanation what kind of interaction occured between starches and the hydrocolloids used by the authors?
A probable interaction was suggested in line 294
Reviewer 5 Report
The manuscript “Physicochemical Properties of Starch Binary Mixtures with Cordia and Ziziphus Gums” reports on a detailed experimental study aimed at assessing the effect of two, rather uncommon, plant gums on the gelatinization and retrogradation behaviors of starches of different origin. The obtained results support the perspective of the employ of the two gums as new natural additives in food.
The writing is clear except for some minor issues I am reporting below:
1. Introduction
page 3 lines 86-94: about Cordia gum uses add ref: Hussain, S.; Mohamed, A.A.; Alamri, M.S.; Ibraheem, M.A.; Qasem, A.A.A.; Shahzad, S.A.; Ababtain, I.A. Use of Gum Cordia (Cordia myxa) as a Natural Starch Modifier; Effect on Pasting, Thermal, Textural, and Rheological Properties of Corn Starch. Foods 2020, 9, 909. https://doi.org/10.3390/foods9070909
from line 100 to line 113 and at line 336: references are addressed by authors’ surnames and publication year instead by numbers as in the rest of the manuscript. Also the line spacing has changed.
line 106 It is not clear what “specific gravity” means referred to a fruit
2. Materials and Methods
Lines 117 and 119 “were purchased from.…”, not for
2.4
Concerning acetylation: report the employed amount of acetic anhydride, see ref. 21, or at least approximately as a range.
2.7 FTIR Analysis:report the material of the ATR accessory crystal, maybe diamond?
Sections 2.8 and 2.12
(TA Instruments, New Castle DE) and not “Castel”
2.12 better “dependence” than “dependency”
3.1 FTIR
Figure 1. The spectrum is represented as Absorbance, not Transmittance %
The spectra of Figure 1 look not very informative: the signal at 1730 characteristic of the acetyl groups are not well evident, at variance with figure 2 of ref. 21 Therefore I suggest to move the FTIR spectra to an eventual Supplementary Information file.
3.6 Gel Texture
line 339 affiliation ?
3.7
the elastic (G`), viscose (G``) moduli
Conclusion
“weak gel” instead of “week gel”
caption of Figure 4: in the plots shear stress (Pa) is reported vs. Shear rate (s-1) and not the opposite
Beside loosely used terms, e.g. line 444 “more elastic property”
Author Response
Reviewer 5: Comments and Suggestions for Authors
The manuscript “Physicochemical Properties of Starch Binary Mixtures with Cordia and Ziziphus Gums” reports on a detailed experimental study aimed at assessing the effect of two, rather uncommon, plant gums on the gelatinization and retrogradation behaviors of starches of different origin. The obtained results support the perspective of the employ of the two gums as new natural additives in food.
The writing is clear except for some minor issues I am reporting below:
- Introduction
page 3 lines 86-94: about Cordia gum uses add ref:Hussain, S.; Mohamed, A.A.; Alamri, M.S.; Ibraheem, M.A.; Qasem, A.A.A.; Shahzad, S.A.; Ababtain, I.A. Use of Gum Cordia (Cordia myxa) as a Natural Starch Modifier; Effect on Pasting, Thermal, Textural, and Rheological Properties of Corn Starch. Foods 2020, 9, 909. https://doi.org/10.3390/foods9070909
The reference was added
from line 100 to line 113 and at line 336: references are addressed by authors’ surnames and publication year instead by numbers as in the rest of the manuscript. Also the line spacing has changed.
References format was changed
line 106 It is not clear what “specific gravity” means referred to a fruit
relative density, is a better term
- Materials and Methods
Lines 117 and 119 “were purchased from.…”, not for
Done
2.4
Concerning acetylation: report the employed amount of acetic anhydride, see ref. 21, or at least approximately as a range.
The amount was about 4 ml
2.7 FTIR Analysis :report the material of the ATR accessory crystal, maybe diamond?
It was diamond
Sections 2.8 and 2.12
(TA Instruments, New Castle DE) and not “Castel”
Corrected
2.12 better “dependence” than “dependency”
Done
3.1 FTIR
Figure 1. The spectrum is represented as Absorbance, not Transmittance %
The spectra of Figure 1 look not very informative: the signal at 1730 characteristic of the acetyl groups are not well evident, at variance with figure 2 of ref. 21 Therefore I suggest to move the FTIR spectra to an eventual Supplementary Information file.
A new graph was added, by expanding the part of the profile of interest, I think the graph is more informative.
3.6 Gel Texture
line 339 affiliation ?
The word is association rather than affiliation.
3.7 the elastic (G`), viscose (G``) moduli
Done
Conclusion
“weak gel” instead of “week gel”
Done
caption of Figure 4: in the plots shear stress (Pa) is reported vs. Shear rate (s-1) and not the opposite
Corrected
Beside loosely used terms, e.g. line 444 “more elastic property”
Done
Round 2
Reviewer 1 Report
This manuscript needs many corrections and English reviews. The attached file of my feedback is just some errors, not all of them. The authors need to check it and modify it carefully if they want to edit the manuscript.
Overall, it's hard to accept this manuscript with many errors.

Author Response
Each and every one of the suggestions was taken into consideration.
Reviewer 2 Report
- With a special thanks to the writers, there is a serious problem with the language of the text. Please double check the structure, grammar and use of the words. In addition, there are some basic rules which should be followed in writing a scientific article, including: order of the paragraphs, font and style of the writing, spacing, numbering all the tables and figures, etc., which have not been applied in the revised version of the article. I strongly suggest a through revise on the whole text. Unfortunately, the innovations and highlights of the scientific work are missing due to language and structure errors.
- The basic information of the extracted gums should be added: a schematic molecular structure, gum composition (protein, fat, ash, etc.), water holding capacity, etc. The latter experiments should be performed on the extracted gums to provide readers with more complete data about the raw materials.
- Please avoid mentioning methods in the introduction and result part, as well as results in the material and method part.
- Please edit the way of reporting the provider companies.
- Numbering of the subtitles should be edited.
- Figures are not located in the related paragraph and sections.
- Ln 300: are there any references?
- It would be better if the application of blending gums in the food industry, in the conclusion part, is further explained.
- Improvement of the references is necessary.
Author Response
The manuscript was corrected for both grammatical and technical/scientific errors.
Round 3
Reviewer 1 Report
Although the authors are patient and modified some errors in the manuscript, the authors just edited the examples I pointed out, not going through the whole manuscript. For example, the basic technical symbols were inconsistent or incorrect (which I already mentioned in the first and second review), but these inconsistencies are still present in the third version, such as storage modulus and loss modulus, the onset temperature and peak temperature. Not only technical symbols, units (line 475, line 161), and equations (for example, Equation (1) between lines 195-196, Equation (2) in line 202) are not correct, maybe because of the typo.
Again, if the authors want to improve the manuscript, go through the whole manuscript to get the consistency for technical symbols, equations, units, plant names first. It’s hard to accept the manuscript with errors for fundamental things.
Author Response
The whole manuscript was checked for consistency on technical symbols, equations, units.
Reviewer 2 Report
Unfortunately, there is no improvement in the language and content of the manuscript after three times revision. The summary of the article, discussion of the results, conclusion, future of the work and particularly the structure of the writing should be fundamentally modified. I strongly suggest the writers to pay more attention to the structure of the scientific writings in their next works.
Author Response
We tried to go over the MS and rewrite a big portion of it, despite the fact that your criticisms were not specific enough to address. If you don't mind, please be more explicit if you have any other comments. I realize you may disagree, but your comments come across as such. I hope we were able to come near to what you had in mind.